# PERK recruits E-Syt1 at ER–mitochondria contacts for mitochondrial lipid transport and respiration

Maria Livia Sassano[1,2], Alexander R. van Vliet[1], Ellen Vervoort[1,2], Sofie Van Eygen[1,2], Chris Van den Haute[3], Benjamin Pavie[4], Joris Roels[4,5], Johannes V. Swinnen[6], Marco Spinazzi[7], Leen Moens[8], Kristina Casteels[9], Isabelle Meyts[8], Paolo Pinton[10], Saverio Marchi[11], Leila Rochin[12], Francesca Giordano[12], Blanca Felipe-Abrio[1,2], and Patrizia Agostinis[1,2]

**The integrity of ER–mitochondria appositions ensures transfer of ions and phospholipids (PLs) between these organelles and exerts crucial effects on mitochondrial bioenergetics. Malfunctions within the ER–mitochondria contacts altering lipid trafficking homeostasis manifest in diverse pathologies, but the molecular effectors governing this process remain ill-defined. Here, we report that PERK promotes lipid trafficking at the ER–mitochondria contact sites (EMCS) through a non-conventional, unfolded protein response-independent, mechanism. PERK operates as an adaptor for the recruitment of the ER–plasma membrane tether and lipid transfer protein (LTP) Extended-Synaptotagmin 1 (E-Syt1), within the EMCS. In resting cells, the heterotypic E-Syt1-PERK interaction endorses transfer of PLs between the ER and mitochondria. Weakening the E-Syt1-PERK interaction or removing the lipid transfer SMP-domain of E-Syt1, compromises mitochondrial respiration. Our findings unravel E-Syt1 as a PERK interacting LTP and molecular component of the lipid trafficking machinery of the EMCS, which critically maintains mitochondrial homeostasis and fitness.**

## Introduction

The ER–mitochondria contact sites (EMCS) are subdomains of smooth ER in close apposition (usually in the range of 10–80 nm) to the mitochondrial outer membrane, which are maintained by dedicated tether, spacer, and lipid transfer proteins (LTPs; Scorrano et al., 2019). EMCS, also termed mitochondria-associated ER membranes (MAMs), are essential regulators of $Ca^{2+}$, metabolites, and phospholipid (PL) transfer between these organelles (Prinz et al., 2020; Vance, 2014). A long appreciated, functional relationship exists between the architectural integrity of EMCS and mitochondrial lipid trafficking (Vance, 1991). Indeed, the lipid composition and structural organization of mitochondrial membranes, depend in large part on the import of essential lipids and precursors from the ER (Ardail et al., 1991; Kornmann et al., 2009; Tamura et al., 2012; Voelker, 1984). While aberrant lipid trafficking between ER and mitochondria is emerging as a persuasive characteristic of aged cells and diseases like neurodegeneration and cancer (Doghman-Bouguerra and Lalli, 2019; Sassano et al., 2017; Wilson and Metzakopian, 2021), a precise understanding of the molecular components regulating

this process is lacking. The expanding physiological roles of EMCS argue that their molecular composition and size are tightly regulated by dynamic recruitment of multifunctional proteins, tethers, and LTPs in order to tailor cellular responses to fluctuating nutritional and stress cues. Assorted ER-resident proteins organize in molecular complexes at the EMCS of mammalian cells (Scorrano et al., 2019). These include PERK (EIF2AK3) and IRE1α, two main effectors of the unfolded protein response (UPR; Fan and Simmen, 2019; Rainbolt et al., 2014; Ron and Walter, 2007; van Vliet and Agostinis, 2018). The UPR is a major transcriptionally regulated adaptive pathway, activated by loss of ER folding capacity following different cellular stresses, such as ER–$Ca^{2+}$ store depletion and glucose deprivation. This results in the accumulation of misfolded proteins in the ER lumen, a condition known as ER stress (Almanza et al., 2019; Ron and Walter, 2007). In particular, the PERK branch of the UPR temporarily halts protein synthesis by phosphorylating eIF2α, while simultaneously increasing the expression of the transcription factor ATF4, which regulates autophagy, amino

[1]Cell Death Research and Therapy Group, Department of Cellular and Molecular Medicine, KU Leuven, Leuven, Belgium;   [2]VIB Center for Cancer Biology, Leuven, Belgium;   [3]Research Group for Neurobiology and Gene Therapy, Department of Neuroscience, Leuven Viral Vector Core, KU Leuven, Leuven, Belgium;   [4]VIB-bioimaging Center UGent, Ghent, Belgium;   [5]Institute for Integrative Biology of the Cell (I2BC), CEA, CNRS, Université Paris-Saclay, Gif-sur-Yvette, France;   [6]Laboratory of Lipid Metabolism and Cancer, Department of Oncology, KU Leuven, Leuven, Belgium;   [7]Neuromuscular Reference Center, CHU Angers, Angers, France;   [8]Laboratory for Inborn Errors of Immunity, Department of Microbiology, Immunology and Transplantation, KU Leuven, Department of Pediatrics, University Hospitals Leuven, Leuven, Belgium;   [9]Woman and Child, Department for Development and Regeneration, KU Leuven, Department of Pediatrics, University Hospitals Leuven, Leuven, Belgium;   [10]Department of Medical Sciences, University of Ferrara, Ferrara, Italy;   [11]Department of Clinical and Molecular Sciences, Marche Polytechnic University, Ancona, Italy;   [12]Inserm, Gif-sur-Yvette, France.

Correspondence to Patrizia Agostinis: patrizia.agostinis@kuleuven.be;   Blanca Felipe-Abrio: blanca.felipeabrio@kuleuven.be.

acid metabolism, oxidative stress, and apoptosis, according to the intensity of ER stress (Harding et al., 2000; Hetz, 2012). Recent studies suggest that in different cellular contexts, PERK maintains mitochondrial homeostasis, mitochondria cristae junctions, and bioenergetics in response to ER stress, through transcriptional-dependent and independent mechanisms. While the molecular underpinning of the former has been confirmed by genetic and pharmacological approaches (Balsa et al., 2019; Chakraborty et al., 2022; Latorre-Muro et al., 2021; Lebeau et al., 2018; Muñoz et al., 2013; Raines et al., 2022), how PERK regulates mitochondrial homeostasis independent of its kinase activity remains elusive. Interestingly, following oxidative ER stress, PERK was shown to prime mitochondrial cell death by tightening the EMCS independent of its kinase activity (Verfaillie et al., 2012). Notwithstanding, whether and how EMCS-associated PERK regulates homeostatic lipid trafficking remains unexplored. This knowledge is particularly relevant considering that the pathogenetic role of PERK mutations in a spectrum of diseases, including the Wolcott-Rallison syndrome (WRS; Julier and Nicolino, 2010), is increasingly linked to mitochondrial dysfunctions and aberrant lipid metabolism (Julier and Nicolino, 2010; Oberhauser and Maechler, 2021), which cannot be explained uniquely by its canonical role in the UPR.

Here, we reveal that PERK promotes lipid transfer between the ER and mitochondria in unstressed cells. We found that this PERK function requires the recruitment of the lipid transfer protein Extended-Synaptotagmin 1 (E-Syt1) through its C2D-C2E domains at EMCS. Using E-Syt1 deletion mutants perturbing PERK-E-Syt1 interaction or ability of E-Syt1 to transport lipids, we portrayed the functional relevance of this axis to preserve mitochondrial lipid homeostasis and respiration.

## Results

### PERK regulates ER–mitochondria phospholipid transfer and mitochondrial lipid homeostasis

Using conventional confocal microscopy, we reported that PERK regulates EMCS under conditions of oxidative ER stress in MEF cells (Verfaillie et al., 2012). Given the limit of the Z resolution of confocal microscopy, we first set out to obtain more accurate 3D images with superior Z-axis resolution of the effects of PERK on EMCS integrity in homeostatic conditions by using Focused Ion Beam Scanning Electron Microscopy (FIB-SEM). For this analysis, we used HeLa cells stably transduced with a control shRNA (shCTR) or shRNA against PERK (shPERK; resulting in ~60% reduction at the protein levels (Fig. S1 A), thus expanding previous data in a different cellular model. 3D reconstitutions of FIB-SEM stacks were used to measure key structural parameters of EMCS—considered as distance equal or less than 25 nm—namely, mitochondrial surface in contact with the ER and EMCS size (Helle et al., 2013; Fig. 1 A, Video 1, Video 2, Video 3, and Video 4). Consistent with previous reports (Hirabayashi et al., 2017; Rizzuto et al., 1998), in shCTR cells, around 20% of mitochondrial surface was in contact with the ER, and this dropped to 10% in shPERK cells (Fig. 1 B). In PERK-silenced cells, the total ER–mitochondria contact area per mitochondria was reduced (Fig. 1 C), while overall mitochondrial morphological parameters

(Fig. 1 D and Fig. S1, B and C) and mass (Fig. S1 D) were unaffected. Thus, PERK contributes to maintain EMCS integrity in unstressed cells.

EMCS constitute the molecular platform for the transport of phosphatidylcholine (PC), phosphatidylinositol (PI), and phosphatidylserine (PS), which are the most abundant PLs in cellular membranes, from the ER where they are synthesized to the mitochondria (Kannan et al., 2017; Tamura et al., 2012; Vance, 2020). Once shuttled to the mitochondria, PS is rapidly converted to phosphatidylethanolamine (PE) by decarboxylation catalyzed by the inner membrane PS decarboxylase (PSD; Vance, 1990).

To decipher the role of PERK in this process, we loaded HeLa cells with 18:1-06:0 N-[7-Nitrobenz-2-oxa-1,3-diazol-4-yl] phosphatidylserine (NBD-PS), a fluorescent analog of PS which can be used to monitor the traffic from ER membranes and subsequently to mitochondrial membranes, where it is rapidly converted into PE (Helle et al., 2013). Live-cell imaging showed that in shCTR cells, NBD-PS redistributed from a main cytoplasmic/ER membrane localization to the mitochondrial network (Fig. 1, E and F) within 30 min, as previously reported (Hailey et al., 2010). In contrast, in shPERK cells, redistribution of NDB-PS to the mitochondria was significantly reduced (Fig. 1, E and F).

We then tested the functional role of PERK in PL trafficking in human fibroblasts derived from a healthy donor and a 2.5-mo-old girl presenting with Wolcott-Rallison Syndrome (WRS) clinically associated with neonatal diabetes and liver dysfunction, caused by a homozygous p.W681X/p.W681X nonsense mutation in *EIF2AK3* (PERK). This resulted in the expression of a PERK truncated variant lacking a large portion of the C-terminal PERK kinase domain (Fig. S1 E). Rapidly after the addition of the fluorescent PS analog, healthy fibroblasts (CTR) showed a clear pattern of mitochondrially redistributed NBD-PS (Fig. 1, G and H). In contrast, in WRS fibroblasts, mitochondria-associated PS signal was significantly diminished (Fig. 1, G and H). Albeit WRS fibroblasts exhibited signs of an altered mitochondrial network, as observed in other pathological conditions caused by chronic loss of PERK signal (Almeida et al., 2022; Lebeau et al., 2018), these results support a role of PERK in ER–mitochondria PL transport.

We next analyzed the conversion of exogenously added NBD-PS to NBD-PE in shCTR and shPERK HeLa cells by one-dimensional thin-layer chromatography (TLC), allowing the separation of PS and PE based on their different polarity (Fig. 2 A; Hölzl and Dörmann, 2021; Watanabe et al., 2020). When calculating the ratio NBD-PE/(NBD-PS + NBD-PE), we found that in PERK silenced cells NBD-PS to NBD-PE conversion was significantly reduced as compared to shCTR cells (Fig. 2 B), further suggesting a role for PERK in ER–mitochondria PS transport.

We then asked whether PERK silencing altered the content of several mitochondrial PLs and/or their precursors, which are regulated by the ER–mitochondria shuttle. To this end, we performed quantitative lipidomics from purified mitochondrial fractions of shCRT and shPERK HeLa cells (Fig. S1 F) and wild-type (PERK$^{+/+}$) and PERK knock out (PERK$^{-/-}$) murine embryonic fibroblast (MEFs; Fig. S1 G). The abundance of PC, PI, PE (PS

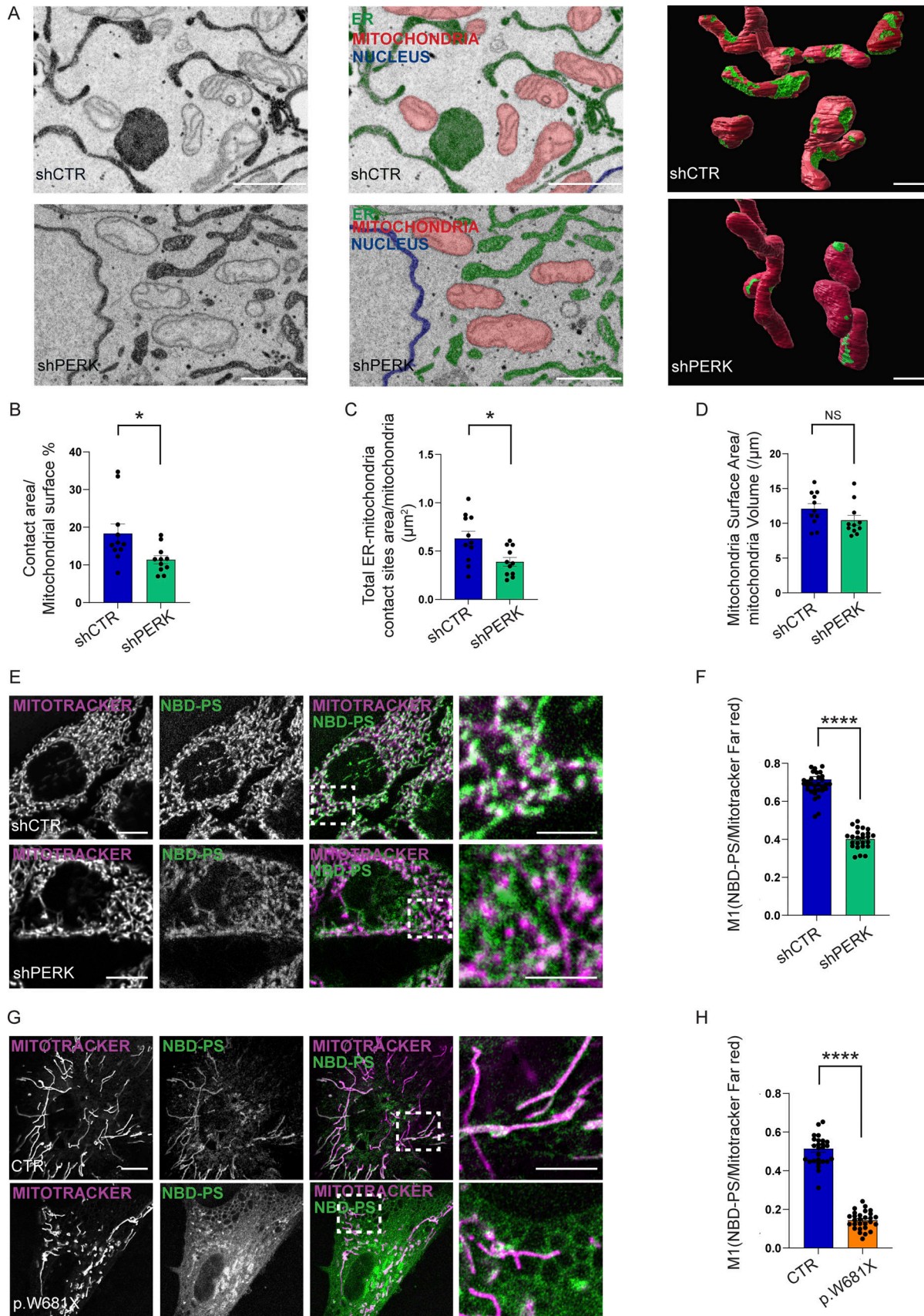

Figure 1. **PERK regulates the integrity of the EMCS and phospholipid trafficking. (A)** Representative electron micrographs and 3D segmentation reconstructed from FIB-SEM image stacks (red = mitochondria, green = ER/EMCS, blue = nucleus) of shCTR and shPERK HeLa cells transiently transfected with

HRP-KDEL-myc. EMCS are indicated by dashed light blue lines (≤25 nm distance between membranes). Scale bar, 1 μm. **(B)** Percentage of contact surface normalized on the total mitochondrial surface/mitochondria. The values plotted are the mean ± SEM from two biological replicates ($n = 11$) analyzed using unpaired Student's $t$ test. **(C)** Total ER–mitochondria contact area/mitochondria. The values plotted are the mean ± SEM from two biological replicates ($n = 11$) analyzed using unpaired Student's $t$ test. **(D)** Average of mitochondrial surface area/volume. The values plotted are the mean ± SEM from two biological replicates ($n = 11$) analyzed using unpaired Student's $t$ test. **(E)** Representative images of NBD-PS co-stained with MitoTracker Far Red in shCTR and shPERK HeLa cells. Scale bar in overview image is 10 μm, and scale bar in magnification is 5 μm. **(F)** NBD-PS and MitoTracker Far Red colocalization analysis of shCTR and shPERK HeLa cells (Manders M1 coefficient). The values plotted are the mean ± SEM from three biological replicates ($n = 37$ and $n = 25$ for shCTR and shPERK, respectively) analyzed using unpaired Student's $t$ test. **(G)** Representative images of NBD-PS co-stained with MitoTracker Far Red in CTR and p.W681X PERK mutant human fibroblasts. Scale bar in overview image is 10 μm, and scale bar in magnification is 5 μm**. (H)** NBD-PS and MitoTracker Far Red colocalization analysis of CTR and p.W681X human fibroblasts (Manders M1 coefficient). The values plotted are the mean ± SEM from three biological replicates ($n = 27$ and $n = 28$ for CTR and p.W681X, respectively) analyzed using unpaired Student's $t$ test. *,$P < 0.05$; ****,$P < 0.0001$; and NS = not significant. See also Video 1, Video 2, Video 3 and Video 4.

is rapidly transformed into PE in the mitochondria), and phosphatidylglycerol (PG; Fig. S1 H) was attenuated in the purified mitochondrial fractions of PERK silenced cells (Fig. 2 C) or PERK$^{-/-}$ cells (Fig. 2 D) as compared to their corresponding PERK proficient counterparts. However, overall mitochondrial PL abundance remained unchanged (Fig. 2, E and F) hinting that other PLs may compensate for the observed reduction in PC, PI, and PE. In line with this, we found an increased trend in the mitochondrial content of diacylglycerol (DG) of shPERK compared to their shCTR (Fig. S1 I). Notably, the overall cellular content of PC, PE, PI, PG, and PS was not grossly changed (Fig. S1 J). Together, these data disclose that PERK deficiency impairs the ER–mitochondrial transport of key phospholipids.

**PERK recruits the lipid transfer protein EXTENDED SYNAPTOTAGMIN-1 at the EMCS**
We next addressed the mechanisms through which PERK regulates mitochondrial lipid homeostasis. PERK silencing did not alter the expression of the mitochondrial PSD nor of the ER-associated PS synthase1/2 (PSS1, PSS2; Fig. S2 A). Our recent PERK interactome studies (Sassano et al., 2021; van Vliet et al., 2017) revealed E-Syt1 as a potential interactor of PERK. E-Syt1 is a member of the conserved E-Syt family of proteins, which in mammals also includes E-Syt2 and E-Syt3. E-Syts are ER-anchored LTPs with a known role in favoring ER-PM tethering (Idevall-Hagren et al., 2015; Kang et al., 2019; Saheki and De Camilli, 2017). We focused on E-Syt1 as a plausible functional PERK interactor given that E-Syt1 is a LTP for which a limited number of interacting partners with ER-PM tethering functions are known (Chang et al., 2013; Kang et al., 2019). Moreover, the possibility that lipid homeostasis could be regulated by a PERK-E-Syt1 interaction at EMCS is unexplored.

We first validated that PERK and E-Syt1 co-immunoprecipitated in HEK-293T cells, by co-transfecting myc-tagged PERK or its kinase-dead mutant (PERK$^{K618A}$) with eGFP-tagged E-Syt1, to control for equal protein expression (Fig. 3 A and Fig. S2 B). Because of the close molecular weight of eGFP-E-Syt1 (146 kD) and myc-tagged PERK (126 kD), both proteins run at a similar height in the gels but were distinguishable as independent bands (Fig. 3 A). A residual GFP-E-Syt1 signal was still visible as upper band in the anti-myc panel, due to spectral bleed-through in our fluorescence-based detection system. PERK and even more the PERK$^{K618A}$ mutant although not significantly (Fig. 3 B) co-immunoprecipitated with eGFP-E-Syt1 (Fig. 3 A). In contrast, the

integral ER protein calnexin (CNX) was detected in the total lysates but was absent in the co-IP, further indicating the specificity of the PERK-E-Syt1 interaction. These results were further confirmed in other co-IP settings using cells expressing eGFP-tagged E-Syt1 followed by GFP-pull down of endogenous PERK (Fig. 3 C).

Thus, PERK independent of its kinase activity interacts, either directly or indirectly, with E-Syt1, confirming our previous Bio-ID data (Sassano et al., 2021).

E-Syt1 has been involved in ER–PM contact formation upon rise in intracellular Ca$^{2+}$ (Idevall-Hagren et al., 2015). We then utilized immunogold electron microscopy (IEM) to visualize whether a fraction of eGFP-E-Syt1 could be found in close proximity with the mitochondria, in resting conditions. Morphological analysis by IEM showed that in HeLa cells around 10% of the total amount of eGFP-E-Syt1 was located at EMCS, a fraction similar to the amount of eGFP-E-Syt1 present at the ER–PM contact sites (Fig. 3, D and E). We then isolated crude mitochondrial fractions (consisting of mitochondria and ER-associated membranes), pure mitochondrial fractions and the EMCS fraction (or MAMs) from shCTR and shPERK HeLa cells (Fig. 3 F) and PERK$^{+/+}$ and PERK$^{-/-}$ MEFs (Fig. S2 C) to analyze the presence of E-Syt1 in these subcellular compartments by immunoblotting. E-Syt1 was found accumulated in the EMCS fraction—while being absent from purified mitochondria as expected—along with PERK, and the known MAM markers Calnexin (CNX) and IP3R3 (Fig. 3 F and Fig. S2 C). At equal EMCS proteins amount, MAM-associated E-Syt1 levels were reduced in absence of PERK (Fig. 3, F and G and Fig. S2, C and D) while this was not observed in total lysates (Fig. 3, F and H; and Fig. S2, C and E), suggesting that PERK may favor E-Syt1 redistribution to EMCS. To corroborate this hypothesis, we then used live-cell imaging in PERK-proficient or depleted Hela cells expressing eGFP-E-Syt1 and visualized mitochondrial network with Mito-Tracker Far Red. In all conditions, eGFP-E-Syt1 signal was distributed as expected within the entire ER network (Giordano et al., 2013). However, while in PERK-proficient cells, a fraction of eGFP-E-Syt1 signal was intermingled with the mitochondrial network; this was reduced in PERK-depleted cells (Fig. 3, J and K). This effect was PERK-specific and independent of its basal kinase activity since reintroduction of the PERK-kinase dead mutant in these cells (Fig. 3 I) rescued this defect to an extent comparable to that of PERK-proficient cells (Fig. 3, J and K). These results suggest that PERK, independently of its

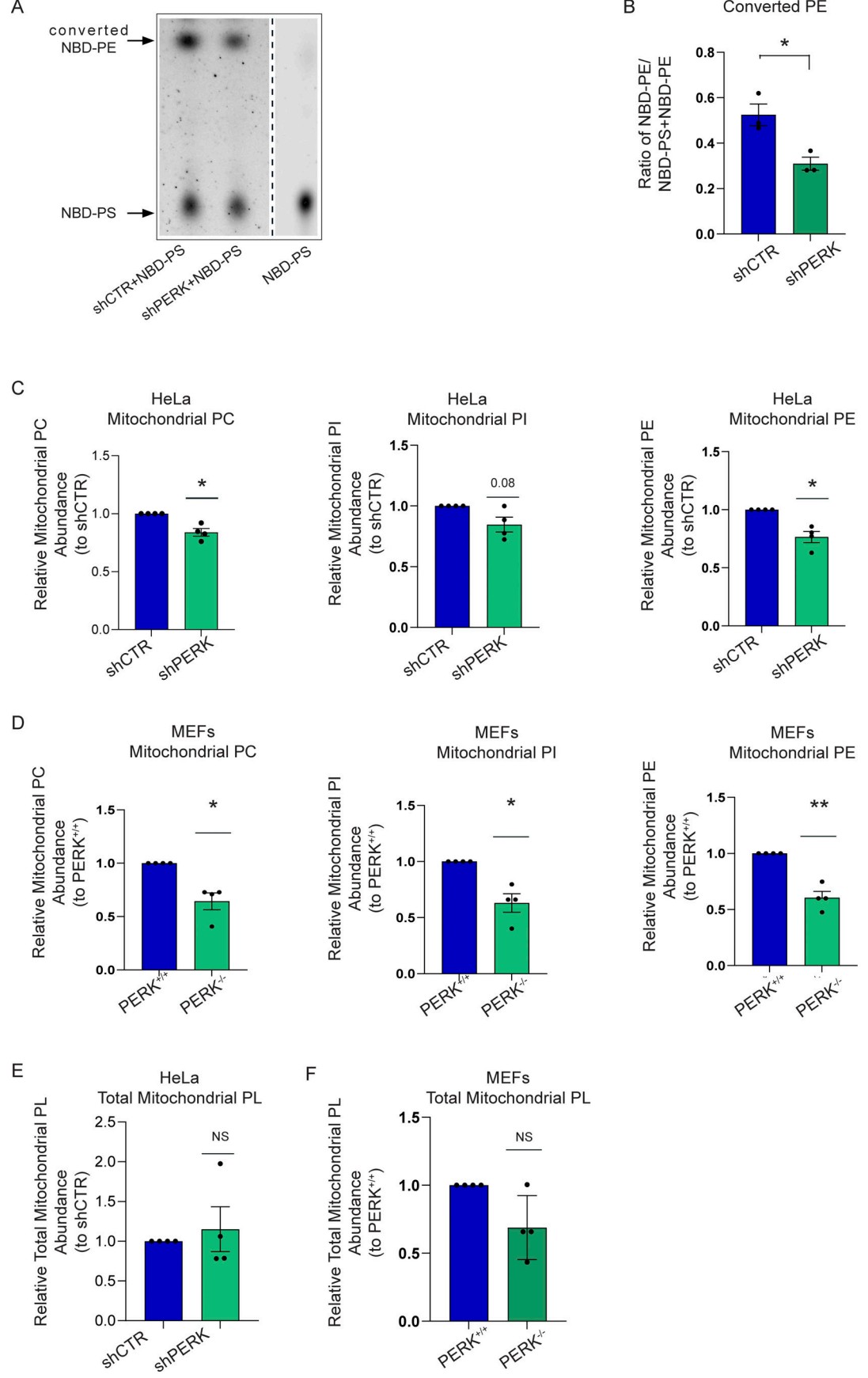

Figure 2.  **PERK regulates mitochondrial lipid homeostasis at the EMCS. (A)** Representative thin-layer chromatography (TLC) image for converted NBD-PE and NBD-PS in shCTR and shPERK cells. Arrows indicate NBD-PS and converted NBD-PE heights. **(B)** Quantification of converted NBD-PE normalized on total NBD-PS + NBD-PE in shCTR and shPERK cells. The values plotted are the mean ± SEM from three biological replicates analyzed using unpaired Student's $t$ test. **(C)** Abundance of PC, PI, PE from purified mitochondrial fractions of shCTR and shPERK HeLa cells, relative to control condition (shCTR). The values plotted are the mean ± SEM from 4 biological replicates analyzed using one sample $t$ test. **(D)** Abundance of PC, PI, PE from purified mitochondrial fractions of PERK$^{+/+}$ and PERK$^{-/-}$ MEF cells, relative to control condition (PERK$^{+/+}$). The values plotted are the mean ± SEM from four biological replicates analyzed using one sample $t$ test. **(E)** Abundance of total mitochondrial phospholipids from purified mitochondrial fractions of shCTR and shPERK HeLa cells, relative to control condition (shCTR). The values plotted are the mean ± SEM from four biological replicates analyzed using one sample $t$ test. **(F)** Abundance of total mitochondrial phospholipids from purified mitochondrial fractions of PERK$^{+/+}$ and PERK$^{-/-}$ MEF cells, relative to control condition (PERK$^{+/+}$). The values plotted are the mean ± SEM from four biological replicates analyzed using one sample $t$ test. *, P < 0.05; **, P < 0.01; and NS = not significant. Source data are available for this figure: SourceData F2.

kinase activity, recruits a fraction of ER-associated E-Syt1 at the intersection between the ER and the mitochondria.

We then investigated whether reducing E-Syt1 expression affected EMCS integrity and mitochondrial lipid composition.

Using orthogonal readouts, we found that in contrast to PERK depletion, reducing E-Syt1 expression did not affect the average number of EMCS per cell (as measured by the PLA assay; Fig. S2, F and I) nor the mitochondrial surface area engaged into EMCS (defined as membrane appositions between the two organelles with <30 nm distance, as measured by HRP-KDEL-myc EM analysis; Fig. S2, J and K). Considering that differences in the number or in the morphology of mitochondria may affect EM analysis of EMCS, we measured the average number of mitochondria per cell and their morphology. Both parameters were unaltered in E-Syt1 silenced cells (Fig. S2, J and K).

Consistent with the observations that optimal Ca$^{2+}$ transfer between the ER and mitochondria occurs through EMCS with a width that lies between 15 and 20 nm (Csordás et al., 2006; Lim et al., 2021), silencing E-Syt1 expression did not elicit alterations in mitochondrial Ca$^{2+}$ transfer (as measured by the mitochondria-targeted protein Aequorin) in response to IP3 generating stimulus ATP (Fig. S2, L–O). In contrast, E-Syt1 silencing largely phenocopied the effects of PERK knockdown on mitochondrial lipid composition. Knocking down E-Syt1 in HeLa cells reduced the mitochondrial abundance of PC, PI, and PE (Fig. 3 L and Fig. S3 A) and PG (Fig. S3 B) without alterations in their global/cellular amount (Fig. S3 C). As observed in PERK-deprived cells, overall mitochondrial PL abundance was unaffected by E-Syt1 silencing (Fig. S3 D), while there was a similar trend toward an increased amount of other mitochondrial PLs such as DG (Fig. S3 E). Moreover, the levels of PSD, PSS1/2 (Fig. S3 F), and the mitochondrial mass (Fig. S3 G) were not affected by E-Syt1 silencing.

**A PERK-E-Syt1 axis mediates phospholipid trafficking at EMCS**
We then set out to gain more insights into the functional effects of E-Syt1–PERK interaction in lipid transfer. E-Syt1 harbors an amino-terminal ER–membrane anchor, a synaptotagmin-like mitochondrial-lipid-binding protein domain (SMP), and five C2 domains: C2A, C2B, C2C, C2D, and C2E, with the C2D-C2E only found in E-Syt1 (Saheki and De Camilli, 2017). Following intracellular Ca$^{2+}$ rise, positively charged C2C domain of E-Syt1 binds PM-rich PI(4,5)P2, favoring ER–PM tethering (Bian et al., 2018; Idevall-Hagren et al., 2015). SMP is a member of the tubular lipid-binding protein (TULIP) superfamily typically

present in proteins associating with membrane contact sites (MCS) with roles in transporting lipids between the ER and other organelles (Prinz et al., 2020; Wong and Levine, 2017).

Our previous Bio-ID interactome analysis did not reveal E-Syt2 as a proximity interactor of PERK (Sassano et al., 2021). However, because ubiquitously expressed E-Syt1 and E-Syt2 share similar structural domains, to further examine the specific role of E-Syt1 as PERK partner, we used E-Syt1/2 double knocked out (DKO) HeLa cells and reconstituted these with eGFP-tagged full length E-Syt1 (E-Syt1FL). eGFP pull-down experiments were performed from isolated crude mitochondrial (mito crude) fractions and showed that endogenous PERK was recovered in eGFP-E-Syt1 expressing DKO cells, but not in cells expressing GFP alone (Fig. 4 A and Fig. S4 A). This suggests that PERK and E-Syt1 interact at the ER–mitochondria interfacing membranes. We then assessed which domain(s) of E-Syt1 was responsible for interaction with PERK. To this end, along with E-Syt1FL, we expressed various eGFP-tagged E-Syt1 deletion mutants, removing either the SMP domain (E-Syt1ΔSMP), the C2C, C2D, and C2E domains (E-Syt1ΔCDE) or the C2D and C2E domains (E-Syt1ΔDE; Fig. 4 B). All these mutants were expressed in DKO cells at a similar extent and comparable to E-Syt1 endogenous levels (Fig. S4, B and C). Co-IP analysis of crude mitochondrial fraction of DKO cells (Fig. 4 C and Fig. S4 D), or from total cell lysates of HEK-293 cells (Fig. S4 E and Fig. S4 F) showed that endogenous PERK was pulled down equally well by the E-Syt1FL or E-Syt1ΔSMP mutant, while expression of the E-Syt1ΔCDE or E-Syt1ΔDE mutants significantly attenuated E-Syt1–PERK interaction (Fig. 4, D–F and Fig. S4G–I). These observations suggest that the cytosolic DE domain of E-Syt1 mediates the association of E-Syt1 with PERK.

We then hypothesized that DE-mediated PERK-E-Syt1 association is required for E-Syt1 localization at the EMCS. DKO cells reconstituted with eGFP-tagged E-Syt1FL or E-Syt1ΔSMP displayed a similar fraction of the eGFP signal intertwined with the mitochondrial network, whereas this colocalization was significantly tempered by PERK-interaction defective E-Syt1ΔDE mutant (Fig. 4, G and H). Congruently, the co-expression of eGFP-tagged E-Syt1FL or E-Syt1ΔDE mutant with STIM1-mCherry (which colocalizes with E-Syt1 in resting conditions and following ER-Ca$^{2+}$ store depletion [Kang et al., 2019]), showed that the PERK-interaction defective mutant displayed an increased ER localization compared to E-Syt1FL (Fig. 4, I and J). Similar results were obtained in PERK$^{+/+}$ MEFs (Fig. S4, J and K), whereas in PERK$^{-/-}$ cells both E-Syt1FL and the E-Syt1ΔDE

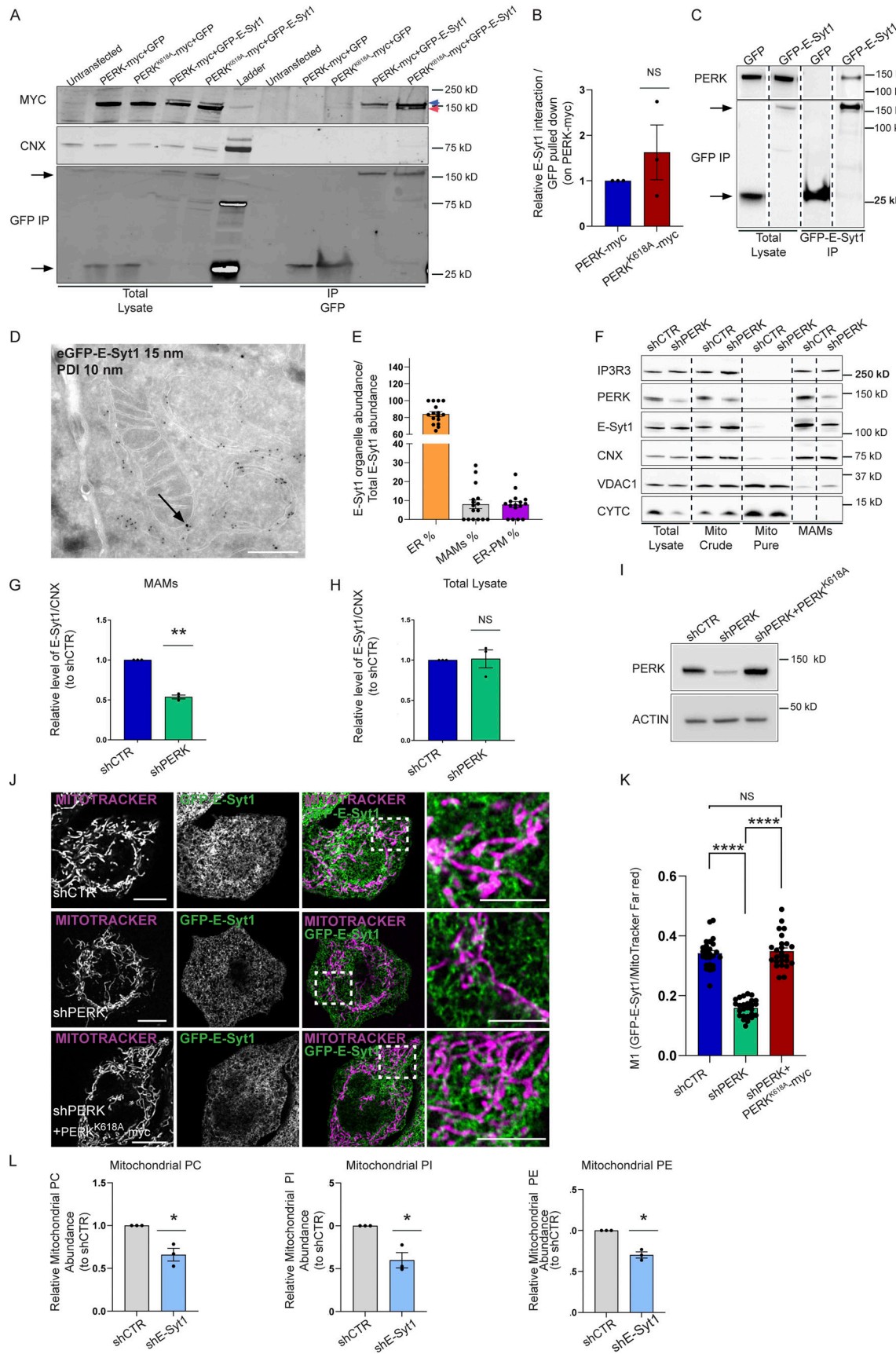

Figure 3. **PERK recruits the lipid transfer protein E-Syt1 at the EMCS. (A)** Representative immunoblot for myc, GFP, and CNX after GFP pull-down showing PERK-E-Syt1 interaction in HEK293-T cells transiently co-transfected with myc-tagged PERK full-length (FL) or myc-tagged PERK kinase dead mutant

(PERKK618A) and with eGFP-empty vector or eGFP-tagged E-Syt1. Untransfected cells are shown as negative control. In the eGFP-E-Syt1 transfected cells, a residual GFP signal is still visible as upper band (blue arrow) above PERK (red arrow) in the anti-myc panel. **(B)** Quantification of the PERK-E-Syt1 interaction normalized on GFP-pulled down and relative to control condition (PERK-myc). The values plotted are the mean ± SEM from three biological replicates analyzed using one sample *t* test. **(C)** Representative immunoblot for eGFP and PERK showing the interaction of PERK and E-Syt1 in HEK293-T cells transiently transfected with eGFP-empty vector or eGFP-tagged E-Syt1. Arrows indicate GFP signals for eGFP-empty vector and eGFP-E-Syt1 pulled down. **(D)** Representative electron micrograph of ultrathin cryosections of HeLa cells transfected with eGFP-E-Syt1 and immunogold stained with anti-GFP (15 nm gold particles) and anti-PDI (10 nm gold particles). Black arrow denotes E-Syt1 detection at the sites of juxtaposition between the ER and the mitochondria membranes, while PDI, a general ER marker, remains in the ER lumen. Scale bar, 500 nm. **(E)** Relative quantification of the cellular distribution of E-Syt1 in the ER, EMCS (MAMs), and plasma membrane (PM). The values plotted are the mean ± SEM (*n* = 16 cellular profiles). **(F)** Representative immunoblot for IP3R3, PERK, E-Syt1, CNX, VDAC1 and CYTC from total lysates, crude mitochondrial fraction (mito crude), purified mitochondrial fraction (mito pure) and MAM fraction of shCTR and shPERK HeLa cells. **(G and H)** Quantification of E-Syt1 level at MAMs (G) and total lysate (H) normalized on CNX levels and relative to control condition (shCTR). The values plotted are the mean ± SEM from three biological replicates analyzed using one sample *t* test. **(I)** Representative immunoblot for PERK in shCTR, shPERK, and shPERK + PERKK618A HeLa cells. ACTIN serves as loading control. **(J)** Representative images from eGFP-E-Syt1 transiently transfected and co-stained with MitoTracker Far Red in shCTR, shPERK and shPERK + PERKK618A HeLa cells. Scale bar in overview image is 10 µm, and scale bar in magnification is 5 µm. **(K)** Colocalization analysis of E-Syt1 and MitoTracker Far Red in shCTR, shPERK and shPERK + PERKK618A HeLa cells (Manders M1 coefficient). The values plotted are the mean ± SEM from three biological replicates (*n* = 26, *n* = 26, and *n* = 25 for shCTR, shPERK, and shPERK + PERKK618A respectively) analyzed using one-way ANOVA, with Tukey's test for multiple comparisons. **(L)** Abundance of PC, PI, PE from purified mitochondrial fractions of shCTR and shE-Syt1 HeLa cells, relative to control condition (shCTR). The values plotted are the mean ± SEM from three biological replicates analyzed using one sample *t* test. *, P < 0.05; **, P < 0.01; ****, P < 0.0001; and NS = not significant. Source data are available for this figure: SourceData F3.

mutant displayed increased ER localization (Fig. S4, J and K). Hence, disruption of the PERK–E-Syt1 interaction or the absence of PERK results in the delocalization of the EMCS-associated fraction of E-Syt1 to the ER.

TLC analysis of NDB-PS to NDB-PE conversion in WT, DKO HeLa cells, and DKO transiently re-expressing E-Syt1 FL (Fig. 5, A and B; and Fig. S5 A) showed that while loss of E-Syt1/2 decreased PE formation compared to the WT cells, re-expression of E-Syt1 could significantly recovered it. Next, we imaged the redistribution of green-fluorescent NBD-PS to the mitochondria (stained by Mitotracker far red) in DKO cells expressing mCherry-tagged E-Syt1FL or E-Syt1ΔCDE (Fig. S5 B), which behaved similarly to the E-Syt1ΔDE (Fig. 4, D and F and Fig. S4, H and I). WT cells displayed a clear pattern of mitochondrially redistributed NBD-PS, while in DKO cells PS trafficking was significantly disrupted (Fig. 5, C and D). Expression of E-Syt1FL in DKO recovered PS trafficking defect (Fig. 5, C and D), whereas that of PERK-interaction defective mutant E-Syt1ΔCDE failed to do so (Fig. 5, C and D). Notably, the expression of the E-Syt1ΔSMP mutant (Fig. 5, E and F), which is required for E-Syt1 PL binding and trafficking (Saheki et al., 2016) and can still bind PERK (Fig. 4 E and Fig. S4 G), also failed to recover the mitochondrial PS defects (Fig. 5, E and F). Thus, E-Syt1 recruitment at EMCS, through its interaction with PERK through C2D-C2E domains, endorses PERK-mediated lipid trafficking.

### The PERK–E-SYT1 axis maintains mitochondrial respiration under steady-state

Alterations of mitochondrial PL composition, and in particular PE abundance, impair formation and/or membrane integration of respiratory chain complexes (Baker et al., 2016; Basu Ball et al., 2018; Joshi et al., 2012; Tasseva et al., 2013) and could have a significant impact on mitochondrial respiration (Mejia and Hatch, 2016). The PERK-eIF2α-ATF4 signal has been shown to regulate mitochondrial metabolism in response to ER stress, but the impact of PERK–E-Syt1 axis on basal oxygen consumption rate (OCR) is unknown.

To address this question and because HeLa cells produce large part of their ATP through the degradation of glucose

(Depaoli et al., 2018), we used conditions that favor oxidative phosphorylation and halt glycolysis. To this end, we replaced the glucose-containing medium with galactose just before measuring OCR. This short-term exposure to galactose did not elicit signs of ER stress (data not shown), which are commonly observed after prolonged glucose deprivation (Iurlaro et al., 2017).

PERK-depleted HeLa cells displayed reduced basal, maximal (uncoupled) respiration, and lower ATP production, which were largely corrected by the re-expression of PERK kinase-dead mutant (Fig. 6, A and B). Consistent with this, WRS-fibroblasts also showed a severely impaired OCR (Fig. 6, C and D) in comparison with healthy fibroblasts whose ability to respire was unaffected by the chemical inhibition of PERK (PKI; Fig. 4 and Fig. 6, C–E).

To evaluate the impact of the PERK–E-Syt1 interaction on oxidative phosphorylation and couple it to the effects observed in PS trafficking (Fig. 5, A–F), we then measured OCR in DKO cells and upon re-expression of E-Syt1FL, E-Syt1ΔDE, or E-Syt1ΔSMP mutants (Fig. S5, C and D). Compared to WT cells, DKO cells exhibited a significantly impaired basal and maximal respiration, and re-expression of E-Syt1FL partially rescued OCR in these cells (Fig. 6, F and G). The observation that E-Syt1 re-expression in DKO HeLa cells does not fully recover the rate of mitochondrial respiration observed in WT cells suggests a possible contribution of E-Syt2 in this process. Notably, expression of the PERK-interaction defective mutant E-Syt1ΔDE or the E-Syt1ΔSMP, which can still bind PERK but is devoid of lipid transfer function, failed to do so (Fig. 6, F and G).

Thus, perturbing the PERK-E-Syt1 interaction or removing the E-Syt1 domain responsible for PL transport (SMP) compromises mitochondrial respiration (Fig. 6 H).

## Discussion

Our study portrays that PERK is a key component of the molecular machinery that maintains homeostatic ER–mitochondria transport of PLs. We further revealed that this function of PERK is favored by heterotypic interaction with the lipid transport protein E-Syt1 at EMCS. Functionally, within the PERK-E-Syt1

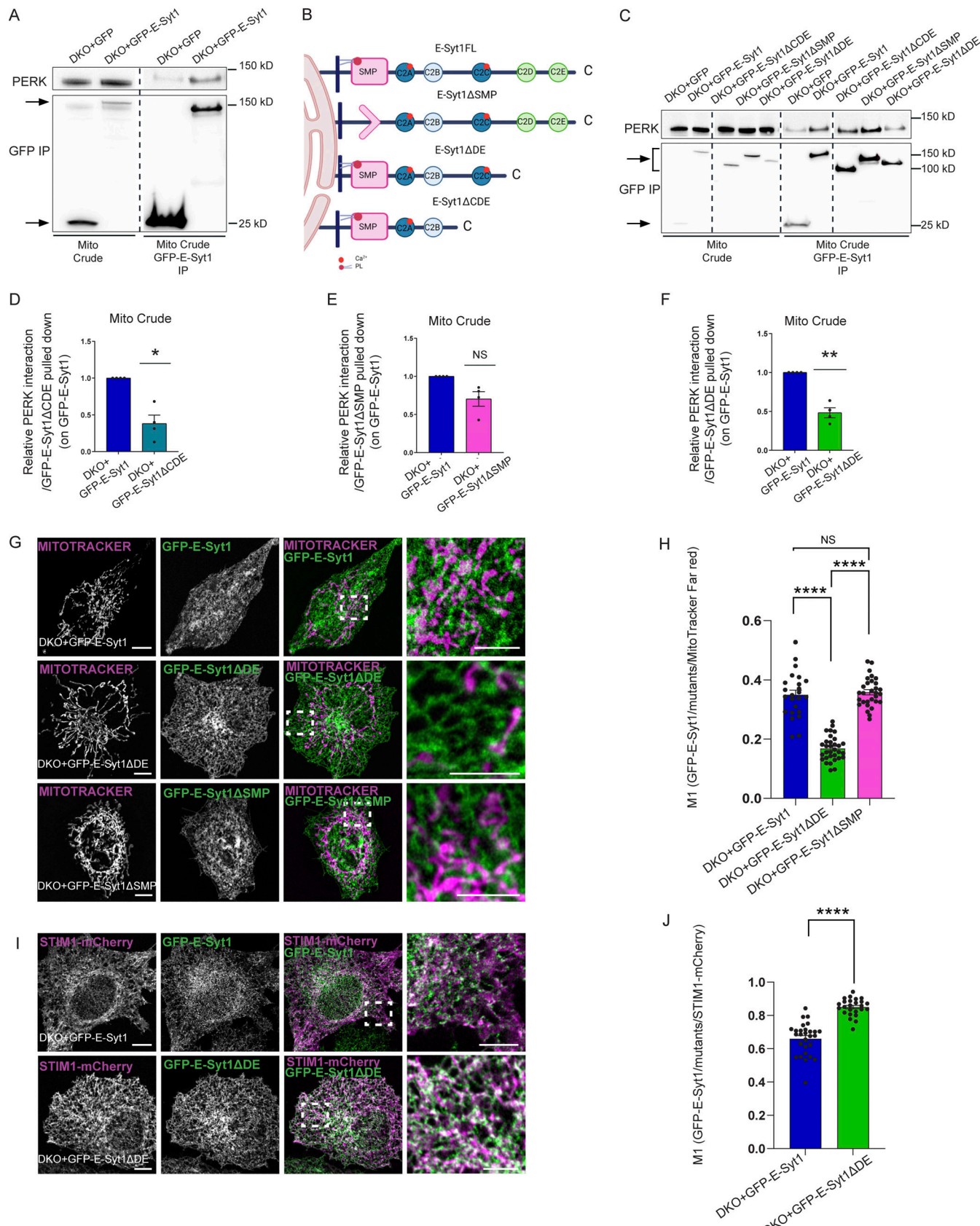

Figure 4. **PERK mediates E-Syt1 localization at the EMCS through E-Syt1 C2D-C2E domain. (A)** Representative immunoblot for eGFP and PERK showing PERK-E-Syt1 interaction from mito crude fractions in DKO HeLa cells transiently transfected with eGFP-empty vector or eGFP-tagged E-Syt1. Arrows indicate GFP signals for eGFP-empty vector and eGFP-E-Syt1 pulled down. **(B)** Schematic representation of the E-Syt1 mutants anchored at the ER membrane used in

this study; E-Syt1 full length (FL), E-Syt1-ΔSMP, E-Syt1-ΔDE, and E-Syt1-ΔCDE. **(C)** Representative immunoblot for eGFP and PERK showing PERK-E-Syt1 interaction from mito crude fractions in DKO HeLa cells transiently transfected with eGFP-empty vector or eGFP-tagged E-Syt1, E-Syt1-ΔCDE, E-Syt1-ΔSMP, and E-Syt1-ΔDE. **(D–F)** Quantification of PERK interaction normalized on E-Syt1-ΔCDE (D), E-Syt1-ΔSMP (E), or E-Syt1-ΔDE (F) eGFP pulled down and relative to control condition (eGFP-E-Syt1). The values plotted are the mean ± SEM from four biological replicates analyzed using one sample $t$ test. **(G)** Representative images from eGFP-E-Syt1 full length, eGFP-E-Syt1-ΔDE or eGFP-E-Syt1-ΔSMP transiently transfected and co-stained with MitoTracker Far Red in DKO HeLa cells. Scale bar in overview image is 10 μm, and scale bar in magnification is 5 μm. **(H)** Colocalization analysis of eGFP-E-Syt1/eGFP-E-Syt1-ΔDE/eGFP-E-Syt1-ΔSMP and MitoTracker Far Red in DKO HeLa cells (Manders M1 coefficient). The values plotted are the mean ± SEM from three biological replicates ($n = 24$, $n = 32$, and $n = 30$ for eGFP-E-Syt1, eGFP-E-Syt1-ΔDE, and eGFP-E-Syt1-ΔSMP respectively) analyzed using one-way ANOVA, with Tukey's test for multiple comparisons. **(I)** Representative images from eGFP-E-Syt1 full length or eGFP-E-Syt1-ΔDE transiently co-transfected with STIM1-mCherry in DKO HeLa cells. Scale bar in overview image is 10 μm, and scale bar in magnification is 5 μm. **(J)** Colocalization analysis of eGFP-E-Syt1/eGFP-E-Syt1-ΔDE and STIM1-mCherry in DKO HeLa cells (Manders M1 coefficient). The values plotted are the mean ± SEM from three biological replicates ($n = 28$ and $n = 25$ for eGFP-E-Syt1 and eGFP-E-Syt1-ΔDE, respectively) analyzed using unpaired Student's $t$ test. *, $P < 0.05$; **, $P < 0.01$; ****, $P < 0.0001$; and NS = not significant. Source data are available for this figure: SourceData F4.

axis, we showed that the expression of E-Syt1 mutants unable to bind PERK fails to rescue the defects in PL transport and mitochondrial respiration observed in E-Syt1/2 DKO cells, while expression of the full length E-Syt1 largely recovers these deficits.

This study reveals that E-Syt1 interacts with PERK through its unique C-terminal C2D-C2E domain and that a fraction of E-Syt1 is found at EMCS in resting cells. Single molecule studies (Bian et al., 2018) indicated that at low Ca²⁺ concentration, the E-Syt1-C2E domain exhibits an intrinsically weak affinity for PM. This may suggest that in resting conditions E-Syt1 may participate in other MCS functions. However, since lipid binding through the SMP domain is modulated by Ca²⁺ (Bian et al., 2018), it is possible, albeit speculative, that local concentration of Ca²⁺ within the microdomains of EMCS (Csordás et al., 2010) may be sufficient to promote SMP PL binding. Alternatively, interaction with PERK may facilitate a conformational change in E-Syt1 releasing charge-based autoinhibitory interaction between the C2A and SMP domains, an interesting hypothesis requiring further structural analysis of PERK–E-Syt1 interaction.

It is remarkable that PERK cooperates with E-Syt1 to regulate lipid trafficking at EMCS. In addition to protein tethers maintaining close contact between the ER and mitochondria, ER-shaping proteins are also required to sustain steady-state levels of PLs (Lahiri et al., 2014; Voss et al., 2012). Tricalbins (yeast orthologs of the mammalian E-Syt1/2/3) have been shown to utilize their curvature-generation and lipid-transport properties to facilitate the formation of cortical ER and ER–PM lipid transport (Collado et al., 2019). Hence, it is possible that membrane curvature and SMP properties of E-Syt1 enable PERK PLs trafficking at EMCS.

However, we cannot exclude that PERK and E-Syt1 interact indirectly and are part of a larger functional complex with other tethers either as separate entities or in partnership with this PERK complex. Consistent with this, severe PL transfer defects are observed in cells missing multiple components of the ER–mitochondria contacts (Lahiri et al., 2014). An interesting partner could be Mitofusin-2 (MFN2), which is a known regulator of PERK (Muñoz et al., 2013) and was recently shown to favor PS transfer to mitochondria to foster PE synthesis (Hernández-Alvarez et al., 2019). These are interesting conjectures that need further exploration.

Recent studies reported that in response to acute ER stress or nutrient deprivation, PERK promotes protective mitochondrial remodeling and mitochondrial respiratory chain supercomplex formation through the coordination of transcriptional (ATF4-SCAF1 dependent) and translational signals (Balsa et al., 2019; Lebeau et al., 2018; Rainbolt et al., 2013). In response to cold stress or beta-adrenergic stimulation in brown adipose tissue, activated/phosphorylated PERK regulates thermogenesis by stimulating biogenesis of respiratory complexes and the MICOS multiprotein complex that controls mitochondria cristae junctions and bioenergetics (Kato et al., 2020; Latorre-Muro et al., 2021). While these studies reveal a role for PERK in key aspects of mitochondrial functions under ER stress conditions, it remained unclear whether PERK exhibited any noncanonical UPR function at EMCS. Although our study provides evidence that PERK regulates PL transfer at EMCS by scaffolding E-Syt1 through mechanisms that do not involve its role as UPR sensor, we cannot exclude that PERK has a complementary role in PL transfer by regulating proximity of these contact sites. The generation of PERK mutants abolishing its interaction with E-Syt1 while preserving PERK's role as regulator of the ER mitochondria contacts could help clarify this point.

However, our finding of a PERK-E-Syt1 axis functioning as a molecular effector of PL transfer between the ER and mitochondria further parses out how a pool of these ER proteins moonlight when present at EMCS to regulate mitochondrial lipid homeostasis and respiration. Indeed, supporting a role of PERK-E-Syt1 axis for PL trafficking and mitochondrial respiration, abolishing either the lipid transfer function of E-Syt1 or disrupting PERK-E-Syt1 interaction cause ER–mitochondria lipid transfer and mitochondrial respiratory capacity defects.

Finally, since mitochondrial and lipid metabolism aberrations are common denominators of devastating diseases like the Wolcott-Rallison syndrome caused by genetic mutations in PERK (*EIF2AK3*; Delépine et al., 2000), our study unravels a crucial new aspect in the growing repertoire of PERK functions.

## Materials and methods

### Cell lines

Simian virus 40 (SV40)-immortalized murine embryonic fibroblasts (MEFs) proficient or deficient for *PERK* (*PERK*⁺/⁺ and *PERK*⁻/⁻ cells, respectively) were a kind gift of David Ron (University of Cambridge, Cambridge Institute for Medical Research, Cambridge, UK; Harding et al., 2000). MEFs cells were maintained in Dulbecco's modified Eagle's medium containing 4.5 g/l

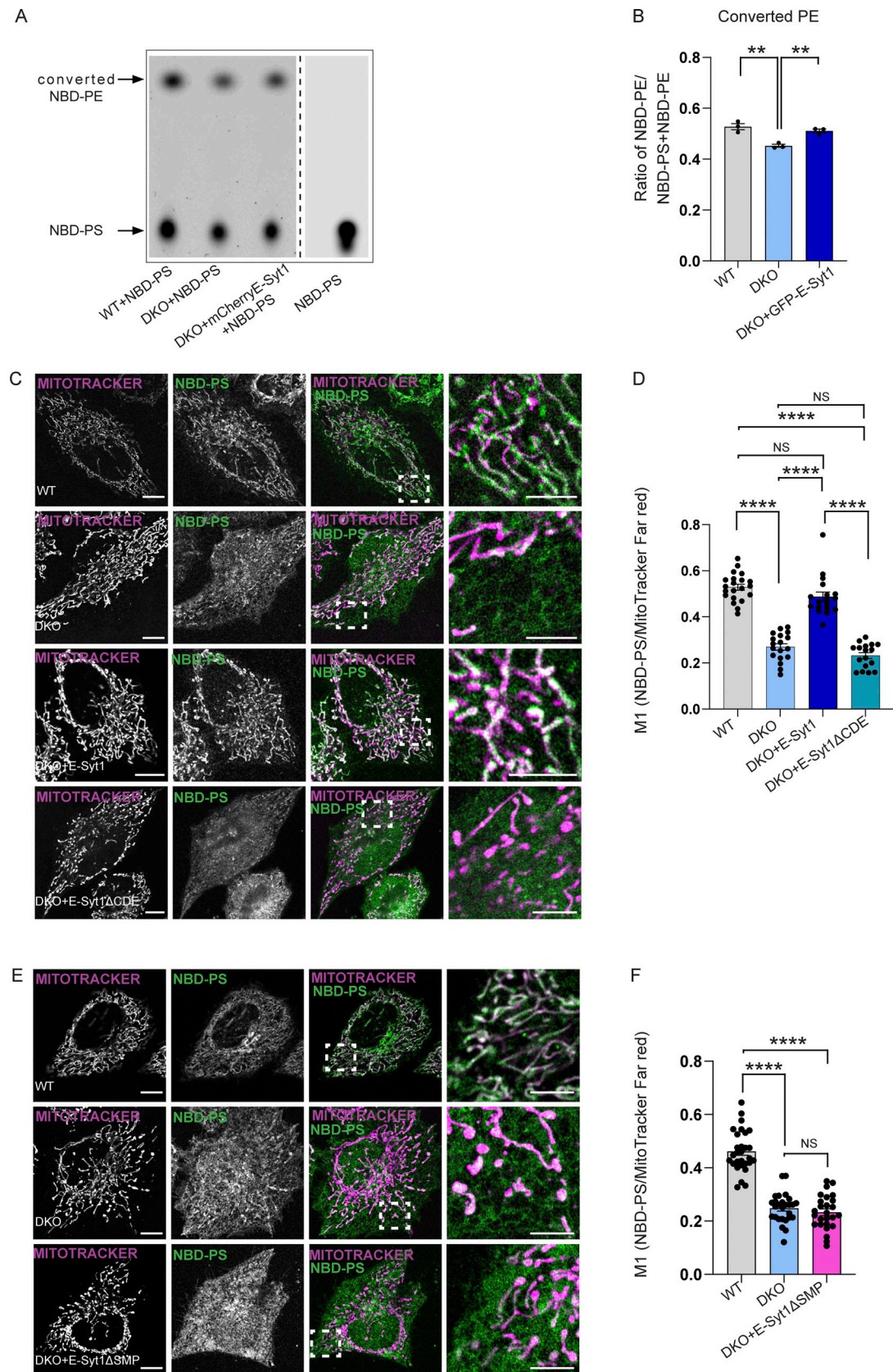

Figure 5. **The PERK-E-Syt1 complex transfers phospholipids at the EMCS. (A)** Representative TLC image for converted NBD-PE and NBD-PS in WT, DKO and DKO cell transiently transfected with mCherry-E-Syt1 full length. Arrows indicate NBD-PS and converted NBD-PE heights. **(B)** Quantification of converted NBD-PE normalized on total NBD-PS + NBD-PE in WT, DKO and DKO + mCherry-E-Syt1 full length cells. The values plotted are the mean ± SEM from three

biological replicates analyzed using one-way ANOVA, with Tukey's test for multiple comparisons. **(C)** Representative images of NBD-PS co-stained with MitoTracker Far Red in WT, DKO HeLa cells, and DKO HeLa cells transiently transfected with mCherry-E-Syt1 full length or mCherry-E-Syt1- ΔCDE. Scale bar in overview image is 10 μm, and scale bar in magnification is 5 μm. **(D)** Colocalization analysis of NBD-PS and MitoTracker Far Red in WT, DKO, DKO + mCherry-E-Syt1 full length (FL), or DKO + mCherry-E-Syt1-ΔCDE HeLa cells (Manders M1 coefficient). The values plotted are the mean ± SEM from three biological replicates ($n = 21$, $n = 18$, $n = 18$, and $n = 17$ for WT, DKO, DKO + mCherry-E-Syt1 FL, or DKO + mCherry-E-Syt1-ΔCDE, respectively) analyzed using one-way ANOVA, with Tukey's test for multiple comparisons. **(E)** Representative images of NBD-PS co-stained with MitoTracker Far Red in WT, DKO HeLa cells and DKO HeLa cells transiently transfected with mCherry-E-Syt1- ΔSMP. Scale bar in overview image is 10 μm, and scale bar in magnification is 5 μm. **(F)** Co-localization analysis of NBD-PS and MitoTracker Far Red in WT, DKO, or DKO + mCherry-E-Syt1-ΔSMP HeLa cells (Manders M1 coefficient). The values plotted are the mean ± SEM from three biological replicates ($n = 29$, $n = 27$, and $n = 26$ for WT, DKO, and DKO + mCherry-E-Syt1-ΔSMP respectively) analyzed using one-way ANOVA, with Tukey's test for multiple comparisons. **, $P < 0.01$; ****, $P < 0.0001$; and NS = not significant. Source data are available for this figure: SourceData F5.

glucose and 0.11 g/l sodium pyruvate and supplemented with 2 mM glutamine, 100 units/ml penicillin, 100 μg/ml strepto-mycin, and 10% fetal bovine serum (FBS). Human cervical cancer (HeLa) cells stably transduced with control vector (shCTR), PERK shRNA (shPERK), control vector (shCTR), and E-Syt1 shRNA (shE-Syt1) were maintained in Dulbecco's modi-fied Eagle's medium containing 4.5 g/l glucose and 0.11 g/l so-dium pyruvate and supplemented with 2 mM glutamine, 100 units/ml penicillin, 100 μg/ml streptomycin, 10% FBS, and 2 μg/ml puromycin. shPERK HeLa cells stably expressing the myc tagged kinase dead mutant (K622A) were maintained in Dulbecco's modified Eagle's medium containing 4.5 g/l glucose and 0.11 g/l sodium pyruvate and supplemented with 2 mM glutamine, 100 units/ml penicillin, 100 μg/ml streptomycin, 10% FBS, 2 μg/ml puromycin and 160 μg/ml hygromycin B. HeLa WT and double knocked out of E-Syt1 and E-Syt2 (DKO) were a kind gift of Pietro De Camilli (Yale University School of Medicine, New Haven, CT, USA; Saheki et al., 2016) and were maintained in Dulbecco's modified Eagle's medium containing 4.5 g/l glu-cose and 0.11 g/l sodium pyruvate and supplemented with 2 mM glutamine, 100 units/ml penicillin, 100 μg/ml streptomycin and 10% FBS. HEK293-T (Veerle Janssens, Laboratory voor Proteïne Fosforylatie en Proteomics, KU Leuven, Leuven, Belgium were maintained in Dulbecco's modified Eagle's medium containing 4.5 g/l glucose and 0.11 g/l sodium pyruvate and supplemented with 2 mM glutamine, 100 units/ml penicillin, 100 μg/ml streptomycin, and 10% FBS.

HeLa cells shCTR and shPERK were already generated in a previous study of our lab (van Vliet et al., 2017). To generate shCTR and shE-Syt1 cells, following viral particle production and transduction, HeLa cells that had efficiently integrated the pLenti shCTR or E-Syt1 shRNA construct (non-targeting control sequence shRNA viral plasmid CTR, Mission shRNA Library, Sigma-Aldrich 5′-CCGGCAACAAGATGAAGAGCACCAACTCGA GTTGGTGCTCTTCATCTTGTTGTTTTT-3′; shRNA viral plasmid against E-Syt1, Mission shRNA Library, Sigma-Aldrich 5′-CCG GTTCACCTAACAGGCCCATATTCTCGAGAATATGGGCCTGTT AGGTGAATTTTTG-3′) were selected with puromycin (2 μg/ml).

To generate shPERK HeLa cells stably expressing the myc tagged kinase dead mutant (K622A), following viral particle production and transduction, HeLa cells that had efficiently in-tegrated the pLenti K622A construct (K622A mutation in the μHsPERK-myc-pUC57 SC1010) were selected with hygromycin B (160 μg/ml).

## Patient-derived cells

Skin-derived fibroblasts (Isabelle Meyts, Laboratory of Inborn Errors of Immunity, KU Leuven, Leuven, Belgium) were ob-tained from a healthy individual (CTR) and a patient with p.W681X/p.W681X nonsense mutation in *EIF2AK3* (PERK) suf-fering from Wolcott-Rallison syndrome. Fibroblasts were iso-lated from a freshly obtained skin biopsy, treated overnight with collagenase Type III (Stemcell, 400 CDU/ml), plated in T25 culture flask in serum-free conditions. After the treatment, the cells were further maintained in Dulbecco's modified Eagle's medium/Nutrient Mixture F-12 containing 4.5 g/l glucose, 0.11 g/l sodium pyruvate and 2 mM glutamine and supplemented with 100 units/ml penicillin, 100 μg/ml streptomycin and 10% FBS.

All cells were maintained in 5% $CO_2$ at 37°C. Cells were routinely checked for mycoplasma contaminations using the Plasmotest kit (Invivogen) according to the manufacturer's instructions.

## Plasmids and cloning

PERK.K622A.9E10.pCDNA myc tagged (21815; Addgene plas-mid), PERK.WT.9E10.pCDNA myc tagged (21814; Addgene plas-mid), eGFP-E-Syt1 (66830; Addgene plasmid) p-eGFP-C3 empty vector plasmid (2489; Addgene plasmid) and pmCherry-C1 empty vector plasmid (3552; Addgene plasmid) were obtained from Addgene. mCherry-E-Syt1, eGFP-E-Syt1-ΔDE, eGFP-E-Syt1-ΔSMP, and mCherry-E-Syt1-ΔCDE were a kind gift from Pietro De Camilli lab (Saheki et al., 2016). KDEL-HRP-myc was obtained from Francesca Giordano (Galmes et al., 2016). eGFP-E-Syt1-ΔCDE plasmid was generated from mCherry-E-Syt1-ΔCDE, E-Syt1-ΔCDE was excised using SalI and SacII restriction en-zymes and cloned into p-eGFP-C3 empty vector (5′-E-Syt1-ΔCDE -SalI-FW cloning primer: 5′-TATATAGTCGACATGGAGCGATCT CCAGGAGA-3′; 5′ E-Syt1- ΔCDE -SacII-Rev cloning primer: 5′-TATATACCGCGGTCGAGGTGGGGCATCCACA-3′). mtAEQ WT was a kind gift from Paolo Pinton (University of Ferrara, Fer-rara, Italy; Robert et al., 2000). pcDNA 3.1 mCherry-STIM1 was a gift from Peter Vangheluwe. Human PERK-myc WT cDNA (μHsPERK-myc-pUC57 SC1010) was generated by GenScript and cloned into a lentiviral transfer plasmid containing ires-hygromycin (pCHMWs-HsPERK-myc-Ires-Hygro SC1017) to se-lect for transduced cells with hygromycin. The K622A mutant was introduced using a gBlock gene fragment (IDT) by replacing a part from the human PERK-myc WT by a fragment that

Figure 6. **The PERK-E-Syt1 axis maintains mitochondrial respiration. (A and B)** OCR of shCTR, shPERK, and shPERK + PERKK618A HeLa cells in galactose media (A); quantification of basal respiration, ATP production and maximal respiration (B). The values plotted are the mean ± SEM from four biological replicates using one-way ANOVA, with Tukey's test for multiple comparisons. **(C and D)** OCR of CTR and p.W681X PERK mutant human fibroblasts in galactose media in untreated conditions and after 2 h pre-treatment with PERK inhibitor (PKI) GSK2606414 1 µM (C); quantification of basal respiration, ATP production and maximal respiration (D). The values plotted are the mean ± SEM from three biological replicates using one-way ANOVA, with Tukey's test for multiple

comparisons. **(E)** Representative immunoblot for PERK and PERK substrate eIf2α (p-eIF2α and total (Tot) EIF2α) in CTR human fibroblasts, respectively in untreated conditions, treated with PERK inhibitor (PKI) GSK2606414 1 µM, ER-stress inducer Thapsigargin (TG) 2 µM for 2 h and TG 2 h + PKI. ACTIN serves as loading control. **(F and G)** OCR of WT HeLa cells electroporated with eGFP, DKO HeLa cells electroporated with eGFP, eGFP-E-Syt1, eGFP-E-Syt1-ΔDE, or eGFP-E-Syt1-ΔSMP in galactose media (E); quantification of basal respiration, ATP production, and maximal respiration (F). The values plotted are the mean ± SEM from five biological replicates using one-way ANOVA, with Tukey's test for multiple comparisons. **(H)** Schematic representation of the effects of E-Syt1 full length (FL), E-Syt1-ΔSMP, and E-Syt1-ΔDE on mitochondrial metabolism at the EMCS. *, $P < 0.05$; **, $P < 0.01$; ***, $P < 0.001$; and NS = not significant. Source data are available for this figure: SourceData F6.

contains the mutation. This construct also allows selection of transduced cells with hygromycin.

## Western blotting

Samples were separated by SDS-PAGE on the Criterion system (Bio-Rad Laboratories) on a 4–12% Bis-TRIS gel and electrophoretically transferred to Protran 2 µm-pored nitrocellulose paper (PerkinElmer). The blots were blocked for 1 h at RT in TBS-T buffer (50 mM Tris, pH 7.4, 150 mM NaCl, 0.1% Tween-20) containing 5% non-fat dry milk and then incubated with selected antibody solutions.

Samples were processed with appropriate secondary antibodies for chemical (HRP-based) detection (Cell Signaling Technologies) or Infrared detection (Thermo Fisher Scientific).

Signal detection was performed using the Typhoon infrared-imaging system (GE Healthcare), Chemidoc TM MP system (Bio-Rad Laboratories) or AI600 Chemiluminescent Imager (GE Healthcare), using ECL solution from Pierce (Thermo Fisher Scientific). Quantifications by densitometry of the bands were calculated using the software Image Studio (Li-Cor Biosciences) or Image Lab 5.2.1 (Bio-Rad Laboratories).

## Antibodies and reagents

Mouse monoclonal anti ACTIN, Sigma-Aldrich, A5441; Mouse monoclonal anti c-myc, Sigma-Aldrich, M4439; Rabbit polyclonal anti Calnexin, Enzo, ADI-SPA-865-F; Mouse monoclonal anti CYTC, BD Bioscience, 556433; Rabbit DyLight 680, Thermo Fisher Scientific, 35569; Mouse DyLight 680 Thermo Fisher Scientific, 35519; Mouse DyLight 800 Thermo Fisher Scientific, 35521; Rabbit DyLight 800 Thermo Fisher Scientific, 35571; Mouse monoclonal anti eIF2α, Cell Signaling, 2103S; Rabbit polyclonal anti-E-Syt1, Sigma-Aldrich, HPA016858; Rabbit polyclonal anti-GFP, Cell Signaling, 2555S; Mouse monoclonal anti-GFP, Life technologies, A11122; HRP Mouse Bioké, Cell Signaling, 7076; HRP Rabbit Bioké, Cell Signaling, 7074; Mouse monoclonal anti IP3R3, BD Bioscience, 610312; Rabbit polyclonal anti-PERK, Cell signaling, 3192S; Rabbit polyclonal anti PERK, Cell signaling, 5683S; Rabbit monoclonal anti Phospho-eIF2α (Ser51), Cell signaling, 3597S; Rabbit polyclonal anti PDI Genetex, GTX30716; Mouse monoclonal anti PSD, Santa Cruz, sc-390070; Rabbit polyclonal anti, PSS1 (B-5), Santa Cruz, sc-515376; Rabbit polyclonal anti PSS2, Sigma-Aldrich, SAB1303408; Rabbit polyclonal VDAC1, Cell Signaling, 4866S; Rabbit polyclonal VDAC1, Abcam, ab15895; Veriblot antibody Abcam, ab131366.

The reagents used were: Antimycin A, Sigma-Aldrich, A8674; Calcium Chloride dihydrate, Sigma-Aldrich, C3881; CHAPS hydrate, Sigma-Aldrich, C3023; Conjugated GFP antibody beads, Laboratory of Chris Ulens; D-Galactose, Sigma-Aldrich, G0750; D-glucose, Sigma-Aldrich, G7021-1KG; DAPI, Thermo Fisher Scientific, 62248; Dulbecco's Modified Eagle's Medium - high glucose, Sigma-Aldrich, D0422; EGTA, AppliChem, A0878; FCCP, Sigma-Aldrich, C2920; Gibco DMEM/F-12, Thermo Fisher Scientific, 11320074; Glucose, Agilent Seahorse, 103577; Glutamine, Sigma-Aldrich, G7513; Glutamine, Agilent Seahorse, 103579; GSK PERK Inhibitor, Toronto Research Company, G797800; Hygromycin B, Invivogen, ant-hg-1; Lipofectamine 2000 Transfection Reagent, Thermo Fisher Scientific, 11668019; MitoTracker FarRed, Thermo Fisher Scientific, M22426; NBD-PS, Avanti Polar Lipids, 810194C; SE Cell Line 4D-Nucleofector X Kit L, V4XC-1024; Oligomycin, Sigma-Aldrich, 75351; Penicillin and streptomycin, Sigma-Aldrich, P0781; Percoll, Sigma-Aldrich, P1644; Pierce ECL Western Blotting Substrate, Thermo Fisher Scientific, 32106X4; Pierce Protein A/G Magnetic Beads, Thermo Fisher Scientific, 88802; Pierce Protease Inhibitor Tablets, EDTA-free, Thermo Fisher Scientific, 88266; Potassium Chloride, Janssen Chimica, 7447407; Protease inhibitor, Thermo Fisher Scientific, A32953; Puromycin, Thermo Fisher Scientific, A11138-03; Protein A/G PLUS-Agarose, Santa Cruz, sc-2003; XF DMEM pH7 7.4, Agilent Seahorse, 103575; Sodium Chloride, Sigma-Aldrich, A0431796; Sodium Pyruvate Solution, Agilent Seahorse, 103578; Sucrose, Acros, A0333146; Thapsigargin, Enzo Life Sciences, BML-PE180; TransIT-X2 Dynamic Delivery System, Mirus Bio, MIR 6000; Tris base, Sigma-Aldrich, 77861; Triton, Sigma-Aldrich, T9234; Tween, Sigma Aldrich, P4780.

## Cell transfection

HEK293T cells were transiently transfected with PERK full length-myc, PERK-K622A -myc, myc-empty vector, eGFP-empty vector, eGFP-tagged E-Syt1, eGFP-E-Syt1-ΔCDE, eGFP-E-Syt1-ΔSMP or eGFP-E-Syt1-ΔDE using Trans-IT X2 transfection reagent accordingly to the manufacturer's instructions. HeLa cells were transiently transfected with human HRP-KDEL-myc, Mitochondrial Aequorin WT, mCherry-empty vector, mCherry-tagged E-Syt1, mCherry-E-Syt1-ΔCDE, eGFP-empty vector, eGFP-E-Syt1, eGFP-E-Syt1-ΔCDE, eGFP-Syt1-ΔSMP or eGFP-E-Syt1-ΔDE using Lipofectamin 2000 (Thermo Fisher Scientific) or electroporated with 4D-Nucleofector (Lonza Bioscience) using SE Cell Line kit (Lonza Bioscience). 24 h after transfections, cells were replated to (microscopy) culture dishes (Mattek corporation) or collected for lysate after 48 h. For Oxygen Consumption Rate (OCR) analysis, cells were plated after nucleofection, and OCR analysis was performed using a Seahorse XF24 (Agilent) Extracellular Flux Analyzer 24 h later.

## Immunoprecipitation (IP)

48 h after transfection with the selected plasmids, cells were collected and lysed at 4°C for 30 min in CHAPS buffer (1% CHAPS, 100 mM KCl, 50 mM Tris HCl, 150 mM NaCl, 1x protease

inhibitor [Pierce Protease Inhibitor Tablets, Thermo Fisher Scientific Inc.]) in MQ water. Cells were centrifuged to remove debris and 500 µg of proteins from the supernatant was combined with primary antibody/conjugated GFP antibody beads overnight (ON) at 4°C. Proteins were eluted with sample buffer (62.5 µM Tris-HCl, 10% glycerol, 2% SDS, 1x protease inhibitor, 1x phosphatase inhibitor (Pierce phosphatase Inhibitor Tablets, Thermo Fisher Scientific Inc.) in MQ water and loaded on gel for Western blot analysis. For developing IPs, ChemiDoc MP Imaging system and Rabbit DyLight 680/Mouse DyLight 800 secondary antibodies were used in cells transiently co-transfected with eGFP-empty vector or eGFP-tagged E-Syt1 and myc-tagged PERK-FL or myc-tagged PERK$^{k618A}$; for the rest of the experiments, HRP-based detection using as secondary antibody the Veriblot antibody was used in order to avoid the detection of the long and short chains of the IgG.

## Ca²⁺ measurements

HeLa cells were transfected with mitochondrial Aequorin WT (mtAEQ WT). After 48 h, cells were incubated with 5 µM coelenterazine for 1.5 h in Krebs–Ringer modified buffer (KRB: 125 mM NaCl, 5 mM KCl, 1 mM Na3PO4, 1 mM MgSO4, 5.5 mM glucose, and 20 mM 4-(2-hydroxyethyl)-1-piperazineethanesulfonic acid [HEPES], pH 7.4, at 37°C) supplemented with 1 mM CaCl₂, and then transferred to the perfusion chamber.

Aequorin measurements were acquired in KRB supplemented with 1 mM CaCl₂, and the agonist was added to the same medium as indicated in figure legends. Cells were then lysed with Triton X-100 in a hypotonic Ca²⁺-rich solution (10 mM CaCl₂ in H2O), thus discharging the remaining aequorin pool. The light signal was collected and calibrated into [Ca²⁺] values using the algorithm reported in Rizzuto et al. (1992).

## HRP detection

HeLa cells expressing HRP-KDEL-myc were fixed on coverslips with 1.3% glutaraldehyde in 0.1 M cacodylate buffer, washed in 0.1 M ammonium phosphate [pH 7.4] buffer for 1 h, and HRP was visualized with 0.5 mg/ml DAB and 0.005% H2O2 in 0.1 M ammonium phosphate [pH 7.4] buffer. Development of HRP (DAB dark reaction product) took between 5 to 20 min and was stopped by extensive washes with cold water. Cells were postfixed in 2% OsO4+1% K3Fe(CN)6 in 0.1 M cacodylate buffer at 4°C for 1 h, washed in cold water and then contrasted in 0.5% uranyl acetate for 2 h at 4°C, dehydrated in an ethanol series and embedded in epon as for conventional EM. Ultrathin sections were counterstained with 2% uranyl acetate and observed under a FEI Tecnai 12 microscope, 4,800X magnification, equipped with a CCD (SiS 1kx1k keenView) camera and iTEM acquisition software.

## Immunogold labeling

HeLa cells were fixed with a mixture of 2% PFA and 0.125% glutaraldehyde in 0.1 M phosphate buffer [pH 7.4] for 2 h and processed for ultracryomicrotomy as described previously (Slot and Geuze, 2007). Ultrathin cryosections were double-immunogold-labeled with antibodies (GFP, Life technologies, A11122, 1:100; PDI, Genetex, GTX30716, 1:500) and protein A

coupled to 10 or 15 nm gold (CMC, UMC Utrecht, The Netherlands), as indicated in the legends to the figures. Immunogold-labeled cryosections were observed under a FEI Tecnai 12 microscope, 6,800X magnification, equipped with a CCD (SiS 1kx1k keenView) camera and iTEM acquisition software.

## FIB-SEM (focused on ion beam-scanning electron microscopy)

HeLa cells expressing HRP-KDEL-myc were grown on gridded coverslips and fixed in freshly prepared fixative (2% paraformaldehyde PFA, Applichem), 2.5% gluteraldehyde (GA, EMS) in 0.1 M sodium cacodylate (Sigma-Aldrich) buffer, pH 7.4 at RT for 30 min. Fixative was removed by washing 5 × 3 min in 0.1 M cacodylate buffer, and subsequently, samples were incubated in 20 mM glycine in 0.1 M sodium cacodylate for 5 min to quench unreacted glutaraldehyde. After another series of washes in 0.1 M sodium cacodylate, HRP in the samples was visualized with freshly prepared 0.5 mg/ml DAB and 0.005% H2O2 in 0.1 M sodium cacodylate. The reaction was followed under a light microscope and stopped when the staining was sufficiently clear by washing in cacodylate buffer. Next, samples were incubated in 1% osmium (OsO4, EMS), 1.5% potassium ferrocyanide (Sigma-Aldrich) in 0.1 M cacodylate buffer for 40 min at RT. After washing in ddH2O for 5 × 3 min, this was followed by overnight incubation at 4°C in 1:3 UAR in H2O (Uranyl Acetate Replacement [UAR], EMS). The next day, UAR was removed by washing in ddH2O for 5 × 3 min. After final washing steps, the samples were dehydrated using ice-cold solutions of increasing EtOH concentration (30, 50, 70, 90%, 2 × 100%), for 3 min each. Subsequent infiltration with resin (Spurr, EMS) was done by first incubating in 50% resin in ethanol for 2 h, followed by at least 3 changes of fresh 100% resin (including one overnight incubation). Next, samples were embedded in fresh resin and cured in the oven at 65°C for 72 h. For FIB-SEM imaging, embedded cells were mounted on aluminum SEM stubs (diameter 12 mm), and samples were coated with ~20 nm of Platinum (Quorum Q150T ES). FIB-SEM imaging was performed using a Zeiss Crossbeam 540 system with Atlas5 software. The Focused Ion Beam (FIB) was set to remove 5-nm sections by propelling Gallium ions at the surface. Imaging was done at 1.5 kV using an ESB (back-scattered electron) detector.

## Proximity Ligation Assay (PLA)

PLA was performed using Duolink In Situ Red Started Kit Mouse/Rabbit (Duolink In Situ Detection Reagents Red, Sigma-Aldrich, DUO92008; Duolink In Situ PLA Probe Anti-Mouse MINUS, Sigma-Aldrich, DUO92004; Duolink In Situ PLA Probe Anti-Rabbit PLUS, Sigma-Aldrich, DUO92004; Duolink In Situ Mounting Medium with DAPI, Sigma-Aldrich, DUO82040) accordingly to the manufacturer's instructions. Briefly, after fixation with 4% PFA and permeabilization with 0.1% Triton, HeLa cells were incubated with primary antibody, VDAC1 (rabbit; ab15895) and IP3R3 (mouse; BD Bioscience 610312), in blocking solution (PBS + 0.1% Triton X-100 + 4% BSA) ON. Secondary antibodies PLUS and MINUS (anti-rabbit and anti-mouse IgG antibodies conjugated with oligonucleotides) were incubated 1:5 in blocking solution for 1 h at 37°C. The ligation solution (ligation buffer 1:5, ligase 1:40 in MQ) was then incubated for 30 min at

37°C. The amplification solution (amplification buffer 1:5, polymerase 1:40 in MQ) was incubated for 1 h 40 min at 37°C. After incubation, slides were mounted with Duolink In Situ Mounting Medium with DAPI. As a negative control, we incubated VDAC1 alone and with both PLUS and MINUS secondary antibodies.

## Subcellular fractionation

Cells were collected and the resulting pellet after centrifugation (600–800 g for 5 min at RT) was resuspended in 5 ml of Starting Buffer 1 (SB1) containing 225 mM mannitol, 75 mM sucrose and 30 mM Tris–HCl, 0.1 mM EGTA, pH 7.4 and homogenized. Unbroken cells and nuclei were removed by centrifugation of the cell homogenate at 600 g for 5 min (at 4°C). The crude mitochondrial fraction (mito crude) was pelleted by centrifuging the supernatant at 7,000 g for 10 min at 4°C while the supernatant contained the ER. The crude mitochondria were resuspended in SB (225 mM mannitol, 75 mM sucrose and 30 mM Tris–HCl, pH 7.4) and after other two sequential centrifugations (7,000 g and 10,000 g for 10 min at 4°C), the obtained pellet was resuspended in 1 ml of mitochondria resuspending buffer MRB (250 mM mannitol, 5 mM HEPES and 0.5 mM EGTA), layered on top of a percoll gradient (30 and 15%) and spun down at 95,000 g for 40 min (using a Beckman ultracentrifuge, rotor SW41), to separate MAMs from the pure mitochondria (mito pure). Further ultracentrifugation (70Ti rotor, 100,000 g for 1 h at 4°C) was required to obtain the pellet of the MAMs fraction.

## Lipidomics

MS-based lipidomics was performed after isolation of the subcellular fractions, using the Lipometrix platform. Samples were processed as previously reported (Talebi et al., 2018). For the identification of the acyl chain composition of intact lipids: 700 µl of homogenized cells were mixed with 800 µl 1 N HCl:CH3OH 1:8 (v/v), 900 µl CHCl3, 200 µg/ml of the antioxidant 2,6-di-tert-butyl-4-methylphenol (BHT; Sigma Aldrich) and 3 µl of SPLASH LIPIDOMIX Mass Spec Standard (#330707; Avanti Polar Lipids). After vortexing and centrifugation, the lower organic fraction was collected and evaporated using a Savant Speedvac spd111v (Thermo Fisher Scientific) at RT and the remaining lipid pellet was stored at –20°C under argon. For lipid analysis, lipid pellets were reconstituted in 100% ethanol. Lipid species were analyzed by liquid chromatography electrospray ionization tandem mass spectrometry (LC-ESI/MS/MS) on a Nexera X2 UHPLC system (Shimadzu) coupled with hybrid triple quadrupole/linear ion trap mass spectrometer (6500 + QTRAP system; AB SCIEX). Chromatographic separation was performed on a XBridge amide column (150 × 4.6 mm, 3.5 µm; Waters) maintained at 35°C using mobile phase A [1 mM ammonium acetate in water-acetonitrile 5:95 (v/v)] and mobile phase B [1 mM ammonium acetate in water-acetonitrile 50:50 (v/v)] in the following gradient: (0–6 min: 0% B → 6% B; 6–10 min: 6% B → 25% B; 10–11 min: 25% B → 98% B; 11–13 min: 98% B → 100% B; 13–19 min: 100% B; 19–24 min: 0% B) at a flow rate of 0.7 ml/min which was increased to 1.5 ml/min from 13 min onwards. SM, CE, CER, DCER, HCER, LCER were measured in positive ion mode with a precursor scan of 184.1, 369.4, 264.4, 266.4, 264.4, and 264.4 respectively. TAG, DAG and MAG were measured in positive ion mode with a neutral loss scan for one of the fatty acyl moieties. PC, LPC, PE, LPE, PG, PI, and PS were measured in negative ion mode by fatty acyl fragment ions. Lipid quantification was performed by scheduled multiple reactions monitoring (MRM), the transitions being based on the neutral losses or the typical product ions as described above. The instrument parameters were as follows: Curtain Gas = 35 psi; Collision Gas = 8 a.u. (medium); IonSpray Voltage = 5500 V and –4,500 V; Temperature = 550°C; Ion Source Gas 1 = 50 psi; Ion Source Gas 2 = 60 psi; Declustering Potential = 60 V and –80 V; Entrance Potential = 10 V and –10 V; Collision Cell Exit Potential = 15 V and –15 V. (The following fatty acyl moieties were taken into account for the lipidomic analysis: 14:0, 14:1, 16:0, 16:1, 16:2, 18:0, 18:1, 18:2, 18:3, 20:0, 20:1, 20:2, 20:3, 20:4, 20:5, 22:0, 22:1, 22:2, 22:4, 22:5 and 22:6 except for TGs which considered: 16:0, 16:1, 18:0, 18:1, 18:2, 18:3, 20:3, 20:4, 20:5, 22:2, 22:3, 22:4, 22:5, 22:6). Peak integration was performed with the MultiQuantTM software version 3.0.3. Lipid species signals were corrected for isotopic contributions (calculated with Python Molmass, 2019.1.1) and were quantified based on internal standard signals and adheres to the guidelines of the Lipidomics Standards Initiative (LSI; level 2 type quantification as defined by the LSI). Data were normalized on the basis of µg of protein amount.

## TLC (thin layer chromatography)

Lipids were extracted using a modified Bligh and Dyer protocol. Briefly, cells were mixed with 1.0 ml of Ultrapure Water (Sigma-Aldrich), 0.9 ml methanol:HCl(1 N; 8:1) and 0.8 ml chloroform. The organic fractions were evaporated under vacuum. Samples and standards were reconstituted in 100 µl methanol:chloroform (1:3). 20 µl of reconstituted samples were spotted on Aluminium backed Silica TLC plates (Sigma-Aldrich) using Hirschmann microcapillary spotters (Sigma-Aldrich) under a gentle stream of nitrogen. The TLC plate was run in a mixture of chloroform/methylacetate/1-Propanol/methanol/0.25% KCl (25:25:25:10:9) as previously described (Watanabe et al., 2020). The plate was imaged using ChemiDoc MP Imaging system at 488 nm wavelength.

## Mitochondrial mass (mitotracker green)

Cells were washed, trypsinized, collected, and then stained with Mitotracker green 30 nM (M7514, Thermo Fisher Scientific) for 30 min. Samples were then washed two times with PBS and processed using BD FACSCanto II. Analysis was performed using FlowJo v10.8.1 software.

## Mitochondrial bioenergetics

Cells were plated in a Seahorse XF24 (Agilent) Extracellular Flux Analyzer culture plate in culture medium overnight. Prior to the assay, cells were switched to the Mitostress assay medium (Seahorse Biosciences) supplemented with 25 mM galactose, 2 mM glutamine and 1 mM sodium pyruvate, following the manufacturer's instructions. OCR during the Mitostress assay were measured with the Seahorse XF24 (Agilent) Extracellular Flux Analyzer (Seahorse Biosciences) following the manufacturer's instructions. Following the Mitostress assay, 1 µM oligomycin, 0.5 µM FCCP (HeLa shCTR, shPERK, shPERK +

PERK[K618A], WT and DKO) or 2.5 µM FCCP (CTR and p. W681X) and 0.5 µM antimycin A diluted in assay medium were the final concentration in the wells. Once the run was finished, all the medium was removed from the well and all the cells were collected in 10 µl of Laemli buffer for BCA-based protein quantification.

### NBD-PS live cells trafficking

After evaporating the chloroform from the solution NBD-PS (810194C, Avanti polar lipids) with N2 gas, NDB-PS was resuspended in ethanol to a concentration of 0.5 M. HeLa cells, were incubated with NBD-PS 1 mM for 30 min. After 30 min, NBD-PS was washed away and cells were incubated with MitoTracker Far Red (M22426 Thermo Fisher Scientific) for 30 min and then imaged 30 min later the NBD-PS staining. Human fibroblasts were incubated with NBD-PS 2.5 µM for 30 min in combination with MitoTracker Far Red and then imaged at $t = 0$. Images were recorded in live cells with Zeiss LSM 880 Airyscan (Pieter Vanden Berghe, University of Leuven, Bioimaging core VIB-KU Leuven, Sebastian Munck).

### Live cell imaging

High resolution confocal images were recorded on a Zeiss LSM 880 – Airyscan, (Cell and Tissue Imaging Cluster [CIC]), ZEN 2.3 SP1 acquisition software, objective 63X, numerical aperture (NA) 1.4 oil, equipped with temperature (37 °C), $CO_2$ and humidity controlled incubator. Cells were imaged in Krebs solution (150 mM NaCl, 5.9 mM KCl, 1.2 mM $MgCl_2$, 11.6 mM HEPES (pH 7.3), 11.5 mM glucose and 1.5 mM $CaCl_2$). Bright-field images were acquired with an Olympus IX73 (Olympus), cellSens Dimension acquisition software, 10X magnification and provided with incubation and $CO_2$ chamber. PLA images were acquired with an Olympus IX73 (Olympus), cellSens Dimension acquisition software and 63X magnification.

### Image processing, analysis, and statistics

For live imaging, confocal images colocalization analysis was performed using ImageJ/Fiji Software, JAcoP plugin, Manders coefficients. The mitochondrial morphology was analyzed with the macro "Mito-morphology" from ImageJ/Fiji Software.

PLA images were analyzed manually with "Threshold" and "Analyze particles" commands from ImageJ/Fiji Software (http://imagej.nih.gov/ij/).

FIB-SEM images were first preprocessed by denoising using the BM4D algorithm (Maggioni et al., 2013). Then, images were first segmented manually with MIB (microscopy image browser), and then the manual segmentation was used to optimize a semi-automatic segmentation analysis. For the automated analysis, an iterative procedure based on U-Net (Falk et al., 2019) and manual correction was employed. First, annotations of the datasets were obtained manually for the mitochondria and semi-automatically for the ER (particularly thresholding, morphological cleaning, and manual correction). These annotations were used to train an initial 2D U-Net (3D volumes are processed slice by slice). Next, the dataset was segmented with this model, followed by small particle removal

and manual correction. The U-Net model was then trained on the three annotated datasets to improve generalization performance and finally applied on the remaining datasets. These datasets were then post-processed by removing small particles and manual correction to obtain the final segmentation of the mitochondria and ER. Segmented images were then imported in Imaris 9.9 and analyzed with Surface-Surface contact plugin. In Imaris, Surface-area and volume were calculated using the Statistic features.

For the quantification of ER–mitochondria contact sites in HRP-stained Epon sections, the total circumference of each mitochondria and the length of the multiple HRP-positive ER segments closely associated (<30 nm) with them were measured by manual drawing using the iTEM software (Olympus), as in Galmes et al. (2016), on acquired micrographs of HeLa cells for each cell profile. Cells were randomly selected for analysis without prior knowledge of transfected plasmid or siRNA. All data are presented as mean (%) ± SEM.

For the quantification of E-Syt1 immunogold labeling on ultrathin cryosections, 300 gold particles were counted in randomly selected cell profiles and assigned to either reticular ER (ER), ER in contact with mitochondria (EMCS) and ER in contact with PM (ER-PM) using the iTEM software. All data are presented as mean (%) ± SEM of three technical replicates.

All data are represented as mean ± SEM. Data distribution was assumed to be normal but this was not formally tested. Statistical significance between two groups was determined by standard unpaired $t$-test with F-testing or one sample $t$ test. Statistical significance between multiple groups was determined by one-way ANOVA to ensure comparable variance, then individual comparisons performed by Tukey's post-hoc test. Analysis was done in Prism v9.0f, GraphPad.

*, $P < 0.05$; **, $P < 0.01$; ***, $P < 0.001$; and ****, $P < 0.0001$, where a P value <0.05 is considered significant.

### Online supplemental material

Fig. S1, related to Fig. 1 and Fig. 2, shows PERK levels in HeLa and fibroblast cell lines, mitochondrial morphology and mass analysis, immunoblots of cellular fractions, and lipidomics data. Fig. S2, related to Fig. 3, shows immunoblots of PSD, PSS1/2 enzymes, immunoblots of cellular fractions, protein quantifications, PLA assays, HRP-KDEL-myc EM analysis, and $Ca^{2+}$ measurements. Fig. S3, also related to Fig. 3, shows immunoblots of cellular fractions, lipidomics data, immunoblots of PSD, PSS1/2 enzymes, and mitochondrial mass analysis. Fig. S4, related to Fig. 4, shows immunoblots of cellular fractions, immunoblots of transiently transfected cell lines with GFP or GFP-tagged E-Syt1 variants, protein quantifications, immunoblots of co-IP analysis, representative microscopy images and colocalization analysis between GFP-Esyt1/GFP-Esyt1ΔDE and STIM1-mCherry. Fig. S5, related to Fig. 5 and Fig. 6, shows representative microscopy images for the transfection efficiency of mCherry or mCherry-tagged E-Syt1 variants, immunoblots and E-Syt1 protein levels of electroporated cells with GFP or GFP-tagged E-Syt1 variants. Video 1, Video 2, Video 3, and Video 4

show 3D videos of segmentations and reconstructions of FIB-SEM analysis.

## Acknowledgments

The authors thank Dr. Pietro De Camilli (Yale School of Medicine) for providing E-Syt1 plasmids and WT and DKO HeLa cells. We thank Joris Van Asselberghs (Research Group for Neurobiology and Gene Therapy, KU Leuven) for producing PERK K622A-myc plasmid and Dr. David Ron for providing PERK$^{+/+}$ and PERK$^{-/-}$ MEFs cells. We thank Dr. Ali Talebi, Dr. Jonas Dehairs, and Frank Vanderhoydonc for the support in the lipidomics and TLC analysis and Dr. Rita La Rovere for the co-IP analysis. We would like to thank the VIB Biomaging Core (VIB-UGENT, Bioimaging core) for training, support and access to the instrument park, in particular Dr. Saskia Lippens and Anneke Kremer.

The Zeiss LSM 880 – Airyscan from the Cell and Tissue Imaging Cluster (CIC), was supported by Hercules AKUL/15/37_GOH1816N and FWO G.0929.15 to Dr. Pieter Vanden Berghe, KU Leuven and the Bioimaging core VIB-KU Leuven, Dr. Sebastian Munck. P.A. is supported by the C1 KU Leuven Consortium InterAction C14/21/095, grants by the Flemish Research Foundation (FWO-Vlaanderen; G0A3320N, G094922N), the EOS MetaNiche consortium no. 40007532 and the iBOF/21/053 ATLANTIS consortium. Open Access funding provided by KU Leuven Libraries.

Author contributions: M.L. Sassano, B. Felipe-Abrio and P. Agostinis. designed the experiments. M.L. Sassano performed the majority of all the experiments presented in the manuscript with the help of E. Vervoort, S. Van Eygen and B. Felipe-Abrio. M.L. Sassano, B. Felipe-Abrio and A.R. van Vliet collected, analyzed the data and interpreted the results. Cloning and stable cell lines were generated by S. Van Eygen and C. Van den Haute. EM microscopy and analysis were performed by F. Giordano and L. Rochin. FIB-SEM analysis was performed by M.L. Sassano with the help of B. Pavie and J. Roels. Lipidomic analysis was carried out by J.V. Swinnen. The isolation and clinical characterization of patient derived fibroblasts was performed by L. Moens, K. Casteels and I. Meyts. Ca$^{2+}$ measurements were performed by M.L. Sassano with the assistance of P. Pinton and S. Marchi. M. Spinazzi helped with analysis of mitochondria metabolism. P. Agostinis and B. Felipe-Abrio supervised the research and with M.L. Sassano interpreted the results. P. Agostinis, M.L. Sassano and B. Felipe-Abrio wrote the manuscript. P. Agostinis oversaw the project and acquired funding. All authors discussed the results and commented on the manuscript.

Disclosures: All authors have completed and submitted the ICMJE Form for Disclosure of Potential Conflicts of Interest. I. Meyts reported "CSL-Behring Grant Paid to Institution. I. Meyts receives funding from the Jeffrey Modell Diagnostic and Research Foundation, paid to Institution." No other disclosures were reported.

Submitted: 2 June 2022

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

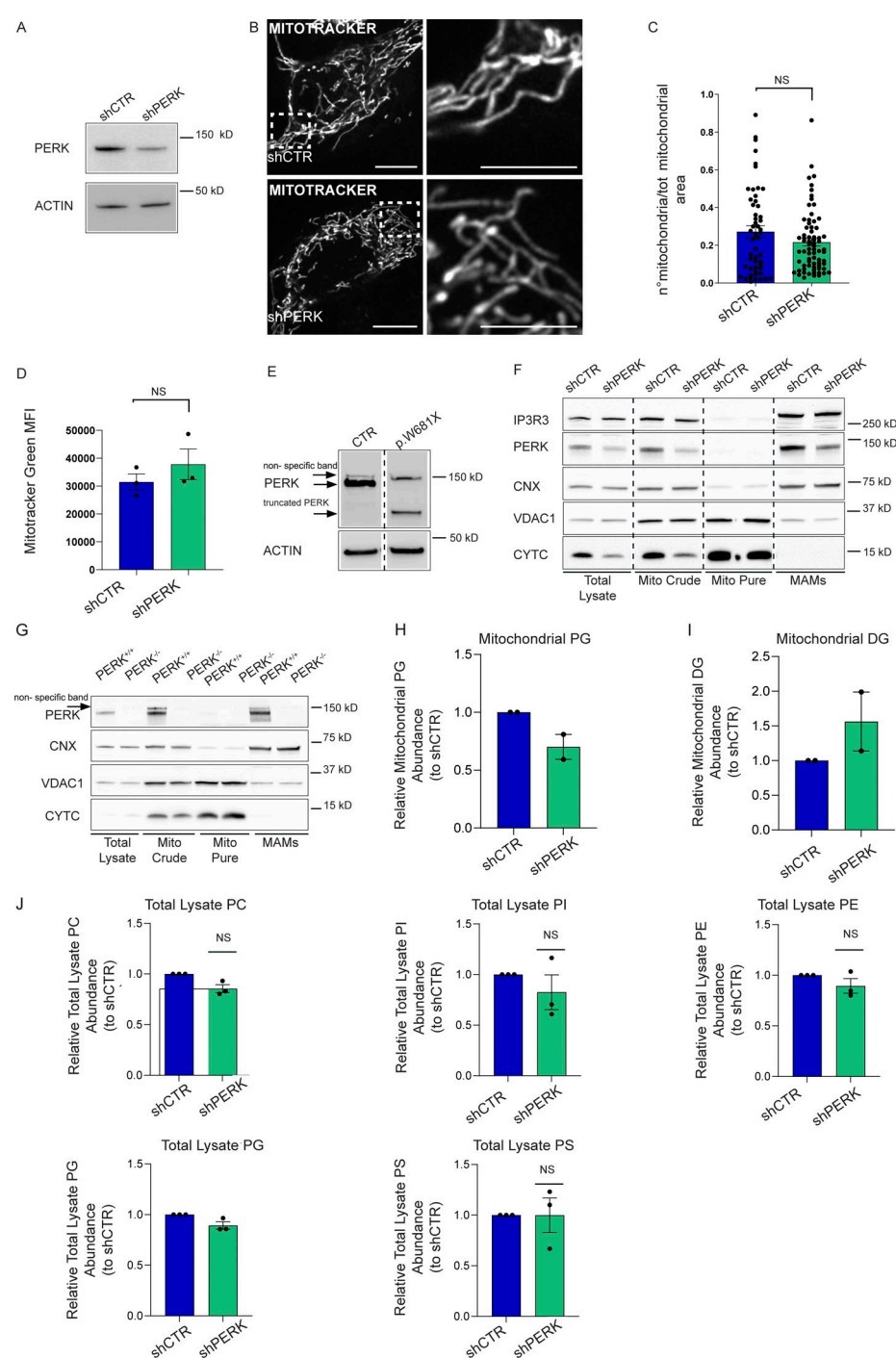

Figure S1. **PERK regulates mitochondrial phospholipid abundance. (A)** Representative immunoblot for PERK in shCTR and shPERK HeLa cells. ACTIN serves as loading control. **(B and C)** Representative images (B) from MitoTracker Far Red staining in shCTR and shPERK HeLa cells; index of mitochondrial fragmentation (C). The values plotted are the mean ± SEM from three biological replicates ($n = 52$ and $n = 72$ for shCTR and shPERK respectively) analyzed using unpaired Student's $t$ test. Scale bar in overview image is 10 μm, and scale bar in magnification is 5 μm. **(D)** Mitotracker Green geometrical mean intensity (MFI) in shCTR and shPERK cells. The values plotted are the mean ± SEM from three biological replicates analyzed using unpaired Student's $t$ test. **(E)** Representative immunoblot for PERK in CTR and p.W681X PERK mutant human fibroblasts. ACTIN serves as loading control. Arrows indicate truncated PERK and a non-specific band. **(F)** Representative immunoblot for IP3R3, PERK, CNX, VDAC1, and CYTC from total lysates, mito crude, mito pure, and MAM fractions of shCTR and shPERK HeLa cells. **(G)** Representative immunoblot for PERK, CNX, VDAC1, and CYTC from total lysates, mito crude, mito pure, and MAM fractions of PERK$^{+/+}$ and PERK$^{-/-}$ MEFs cells. Arrow indicates a non-specific band. **(H)** Abundance of PG from purified mitochondrial fractions of shCTR and shPERK HeLa cells, relative to control condition (shCTR). The values plotted are the mean ± SEM from two biological replicates. **(I)** Abundance of DG from purified mitochondrial fractions of shCTR and shPERK HeLa cells, relative to control condition (shCTR). The values plotted are the mean ± SEM from two biological replicates. **(J)** Abundance of PC, PI, PE, PG, PS from total cell lysates of shCTR and shPERK HeLa cells, relative to control condition (shCTR). The values plotted are the mean ± SEM from three biological replicates analyzed using one sample $t$ test. NS = not significant. Source data are available for this figure: SourceData FS1.

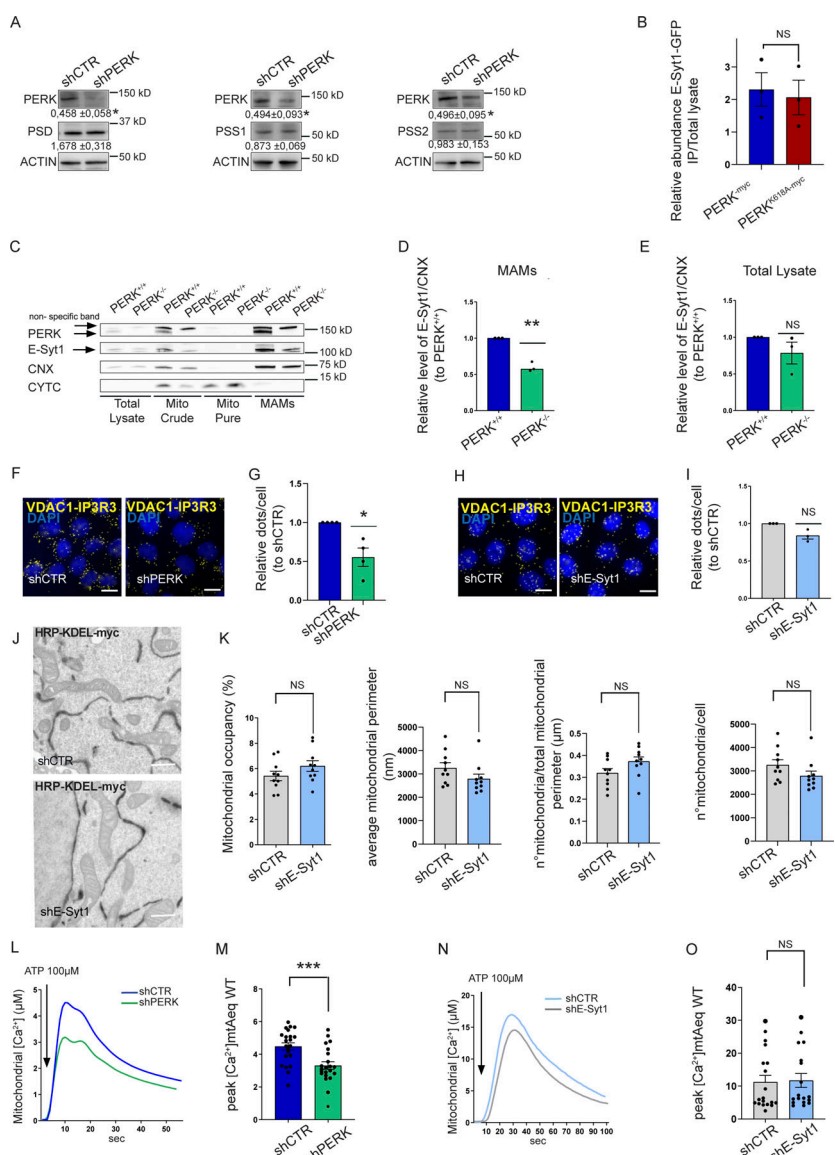

Figure S2. **PERK modulates the proximity and the mitochondrial Ca²⁺ uptake at the EMCS but E-Syt1 does not. (A)** Representative immunoblot for PERK, PSD, PSS1, PSS2 in shCTR and shPERK HeLa cells and quantification of PSD, PSS1, PSS2, and PERK, normalized on ACTIN (loading control) and relative to control condition (shCTR). The values shown are the mean ± SEM from three biological replicates analyzed using one sample *t* test. **(B)** Abundance of GFP-E-Syt1 pulled-down normalized on abundance of GFP-E-Syt1 in total lysate in HEK293-T cells transiently co-transfected with GFP-E-Syt1 and myc-tagged PERK full-length (FL) or myc-tagged PERK kinase dead mutant (PERK^K618A). The values plotted are the mean ± SEM from three biological replicates analyzed using unpaired Student's *t* test. **(C)** Representative immunoblot for PERK, E-Syt1, CNX, and CYTC from total lysates, mito crude, mito pure, and MAMs fractions of PERK⁺/⁺ and PERK⁻/⁻ MEFs cells. Arrows indicate E-Syt1, PERK, and a non-specific band. **(D and E)** Quantification of E-Syt1 levels at MAMs (D) and in the total lysate (E) normalized on CNX levels and relative to control condition (PERK⁺/⁺). The values plotted are the mean ± SEM from three biological replicates analyzed using one sample *t* test. **(F and G)** Representative images (F) in situ PLA in shCTR and shPERK HeLa cells and quantification (G) of number of dots corresponding to IP3R3-VDAC1 interaction per nucleus and relative to control condition (shCTR). The values plotted are the mean ± SEM from three biological replicates analyzed using one sample t test. Scale bar, 10 μm. **(H and I)** Representative images (H) in situ PLA in shCTR and shE-Syt1 HeLa cells and quantification (I) of number of dots corresponding to IP3R3-VDAC1 interaction per nucleus and relative to control condition (shCTR). The values plotted are the mean ± SEM from three biological replicates analyzed using one sample t test. Scale bar, 10 μm. **(J and K)** Representative images (J) from EM analysis in shCTR and shE-Syt1 HeLa cells transfected with HRP-KDEL-myc and quantifications (K) of % mitochondria surface engaged into ER–mitochondria contact sites width below 30 nm (mitochondria occupancy %); mitochondrial average perimeter, number of mitochondria normalized on total mitochondrial perimeter and average number of mitochondria. The values plotted are the mean ± SEM (*n* = 10 cells) analyzed using unpaired Student's *t* test. Scale bar, 500 nm. **(L and M)** Representative traces (L) of mitochondrial calcium uptake after ER calcium depletion (ATP 100 μM) in shCTR and shPERK transiently transfected with mitochondrial Aequorin WT (mtAeqWT); quantification (M) of the mitochondrial peak of [Ca²⁺]. The values plotted are the mean ± SEM from four biological replicates (*n* = 22) analyzed using unpaired Student's *t* test. Arrow indicates the addition of ATP. **(N and O)** Representative traces (N) of mitochondrial Ca²⁺ uptake after ER calcium depletion (ATP 100 μM) in shCTR and shE-Syt1 transiently transfected with mtAeqWT; quantification (O) of the mitochondrial peak of [Ca²⁺]. The values plotted are the mean ± SEM from three biological replicates (*n* = 19 and *n* = 18 for shCTR and shE-Syt1, respectively) analyzed using unpaired Student's *t* test. Arrow indicates the addition of ATP. *, P < 0.05; **, P <0.01; ***, P <0.001; and NS = not significant. Source data are available for this figure: SourceData FS2.

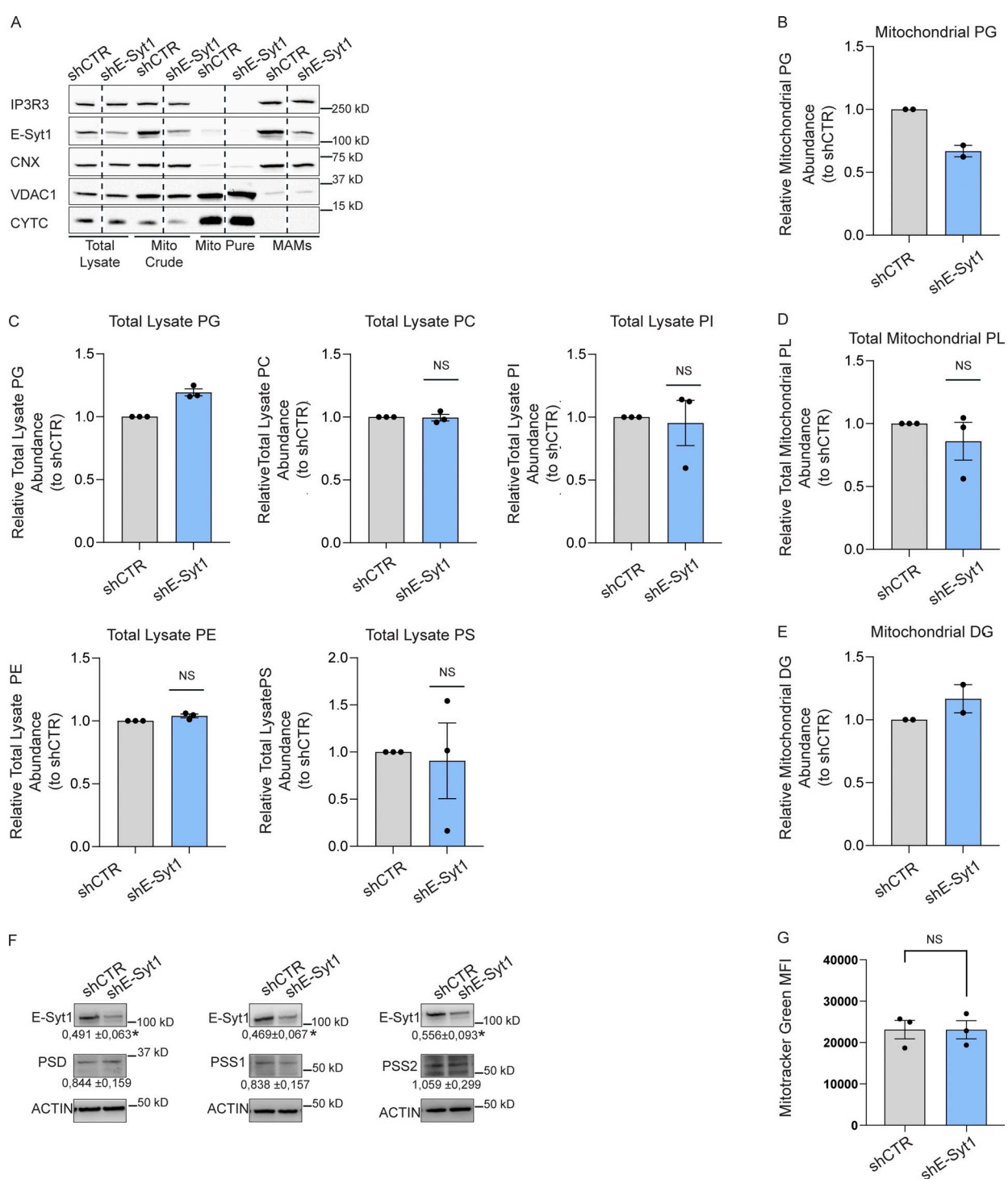

Figure S3.  **E-Syt1 is found at the EMCS and regulates mitochondrial phospholipid abundance. (A)** Representative immunoblot for IP3R3, E-Syt1, CNX, VDAC1, and CYTC from total lysates, mito crude, mito pure, and MAM fractions of shCTR and shE-Syt1 HeLa cells. **(B)** Abundance of PG from purified mitochondrial fractions of shCTR and shE-Syt1 HeLa cells, relative to control condition (shCTR). The values plotted are the mean ± SEM from two biological replicates. **(C)** Abundance of PG, PC, PI, PE, PS from total cell lysates of shCTR and shE-Syt1 HeLa cells, relative to control condition (shCTR). The values plotted are the mean ± SEM from three biological replicates analyzed using one sample *t* test. **(D)** Abundance of total mitochondrial phospholipids from purified mitochondrial fractions of shCTR and shE-Syt1 cells, relative to control condition (shCTR). The values plotted are the mean ± SEM from three biological replicates analyzed using one sample *t* test. **(E)** Abundance of DG from purified mitochondrial fractions of shCTR and shE-Syt1 HeLa cells, relative to control condition (shCTR). The values plotted are the mean ± SEM from two biological replicates. **(F)** Representative immunoblot for E-Syt1, PSD, PSS1, PSS2 in shCTR and shE-Syt1 HeLa cells and relative quantifications of PSD, PSS1, PSS2, and E-Syt1, normalized on ACTIN (loading control) and relative to control condition (shCTR). The values shown are the mean ± SEM from three biological replicates analyzed using one sample *t* test. **(G)** Mitotracker Green geometrical mean intensity (MFI) in shCTR and shE-Syt1 cells. The values plotted are the mean ± SEM from three biological replicates analyzed using unpaired Student's *t* test. NS = not significant. Source data are available for this figure: SourceData FS3.

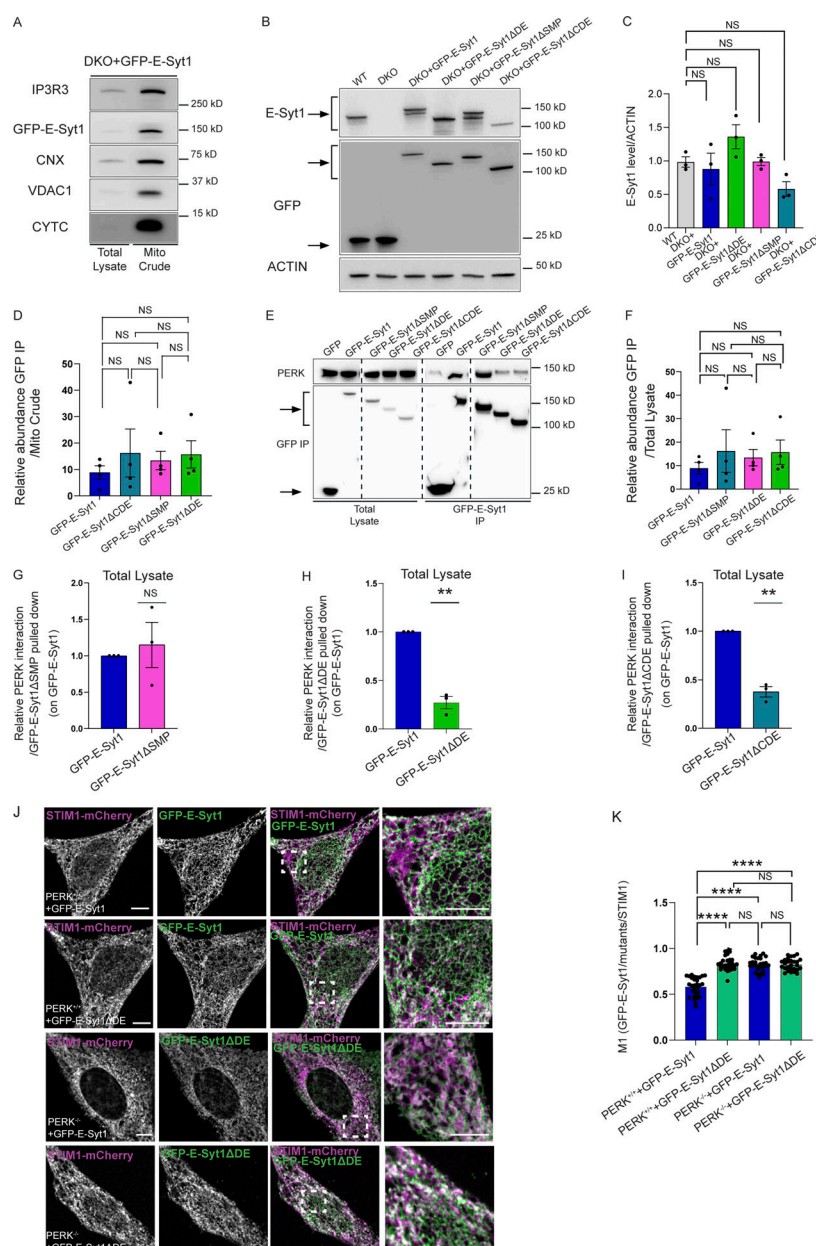

Figure S4. **E-Syt1 interacts through its C2D-C2E domain with PERK. (A)** Representative immunoblot for IP3R3, GFP, PERK, CNX, VDAC1, and CYTC from total lysate and mito crude fractions of DKO HeLa cells transiently transfected with eGFP-E-Syt1. **(B and C)** Representative immunoblot (B) for E-Syt1 and GFP in WT HeLa cells transiently transfected with eGFP and DKO HeLa cells transiently transfected with eGFP, eGFP-E-Syt1, eGFP-E-Syt1-ΔDE, eGFP-E-Syt1-ΔSMP, eGFP-E-Syt1-ΔCDE; quantification (C) of E-Syt1 expression levels normalized on ACTIN (loading control) in WT, DKO + E-Syt1, DKO + E-Syt1-ΔDE, DKO + E-Syt1-ΔSMP, or DKO + E-Syt1-ΔCDE HeLa cells. The values plotted are the mean ± SEM from three biological replicates using one-way ANOVA, with Tukey's test for multiple comparisons. Arrows indicate GFP signals for eGFP-empty vector and eGFP-E-Syt1 mutants transfected and E-Syt1 signals for E-Syt1 endogenous and eGFP-E-Syt1 mutants transfected. **(D)** Abundance of GFP pulled-down normalized on abundance of GFP in mito crude fraction in DKO HeLa cells transiently transfected with eGFP-tagged E-Syt1, E-Syt1-ΔCDE,E-Syt1-ΔSMP, and E-Syt1-ΔDE. The values plotted are the mean ± SEM from four biological replicates analyzed using one-way ANOVA, with Tukey's test for multiple comparisons. **(E)** Representative immunoblot for GFP and PERK showing PERK-E-Syt1 interaction from total lysate in HEK293-T cells transiently transfected with eGFP-empty vector or eGFP-tagged E-Syt1, E-Syt1-ΔSMP, E-Syt1-ΔDE, and E-Syt1-ΔCDE. Arrows indicate GFP signals for eGFP-empty vector and eGFP-E-Syt1 mutants pulled down. **(F)** Abundance of GFP pulled-down normalized on abundance of GFP in total cell lysates in HEK293-T cells transiently transfected with eGFP-tagged E-Syt1, E-Syt1-ΔSMP, E-Syt1-ΔDE, and E-Syt1-ΔCDE. The values plotted are the mean ± SEM from biological replicates analyzed using one-way ANOVA, with Tukey's test for multiple comparisons. **(G–I)** Quantification of PERK interaction normalized on E-Syt1-ΔSMP (G), E-Syt1-ΔDE (H), or E-Syt1-ΔCDE (I) eGFP-pulled down and relative to control condition (eGFP-E-Syt1). The values plotted are the mean ± SEM from three biological replicates analyzed using one sample $t$ test. **(J)** Representative images from eGFP-E-Syt1 full length or eGFP-E-Syt1-ΔDE transiently co-transfected with STIM1-mCherry in PERK$^{+/+}$ and PERK$^{-/-}$ MEFs cells. Scale bar, 10 μm. **(K)** Colocalization analysis of eGFP-E-Syt1/eGFP-E-Syt1-ΔDE and STIM1-mCherry in PERK$^{+/+}$ and PERK$^{-/-}$ MEFs cells (Manders M1 coefficient). The values plotted are the mean ± SEM from three biological replicates ($n = 30$, $n = 29$, $n = 30$, and $n = 27$ for PERK$^{+/+}$+E-Syt1-GFP, PERK$^{+/+}$+E-Syt1ΔDE, PERK$^{-/-}$ + E-Syt1-GFP, and PERK$^{-/-}$ + E-Syt1ΔDE, respectively) analyzed using one-way ANOVA, with Tukey's test for multiple comparisons. **, $P < 0.01$; ****, $P < 0.0001$; and NS = not significant. Source data are available for this figure: SourceData FS4.

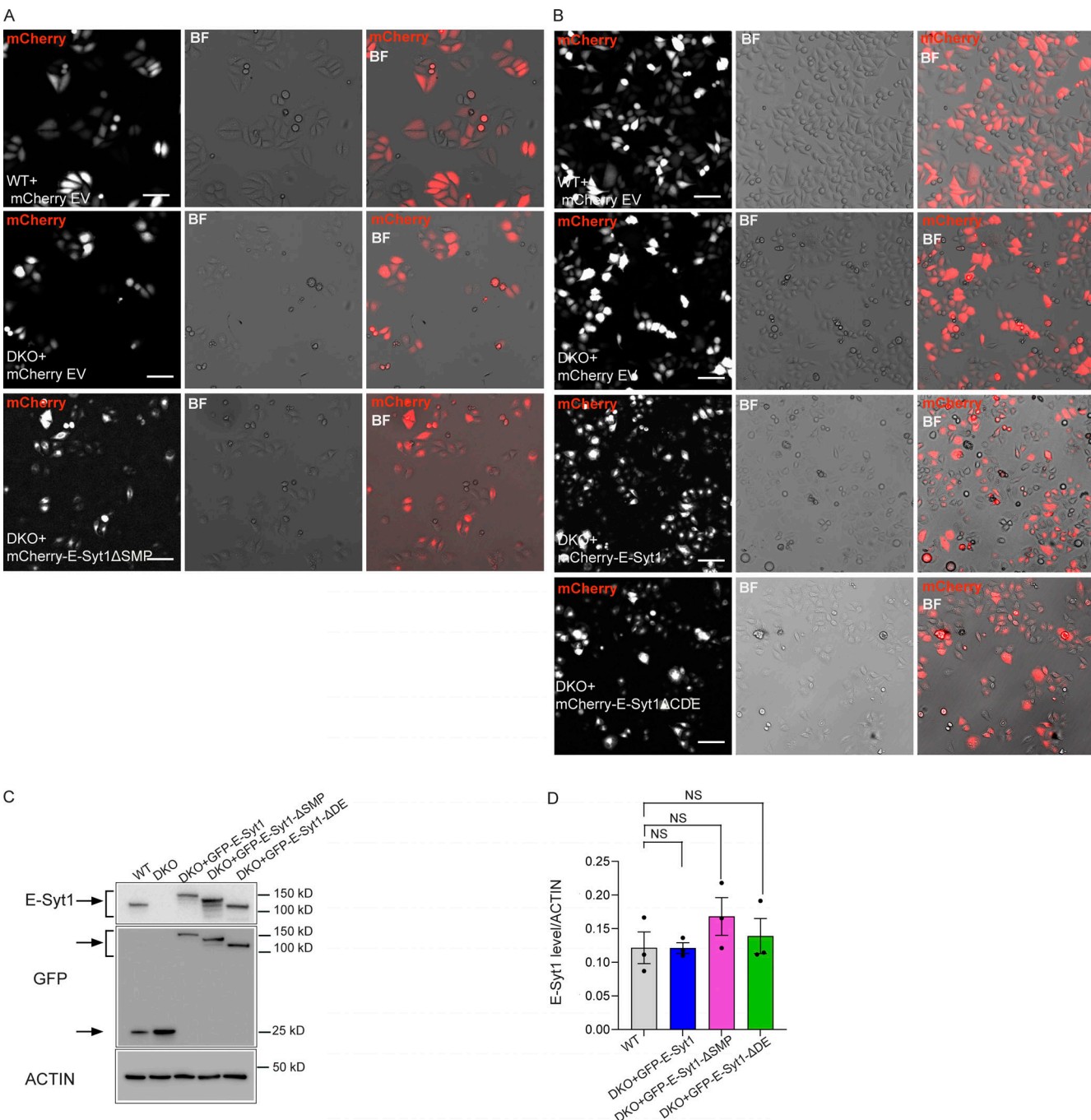

Figure S5. **PERK boosts the OCR independently of its ER stress activity but through E-Syt1 interaction. (A)** Representative images showing the transfection efficiency (shown as merge of mCherry and BF channel) in WT and DKO HeLa cells transfected with mCherry empty vector (EV), DKO with mCherry-E-Syt1- ΔSMP and used in Fig. 5, E and F. Scale bar, 100 μm. **(B)** Representative images showing the transfection efficiency (shown as merge of mCherry and BF channel) in WT and DKO HeLa cells transfected with mCherry empty vector (EV), DKO with mCherry-E-Syt1, or mCherry-E-Syt1- ΔCDE and used in Fig. 5, C and D. Scale bar, 100 μm. **(C)** Representative immunoblot for E-Syt1 and GFP in WT HeLa cells electroporated with eGFP and DKO HeLa cells electroporated with eGFP, eGFP-E-Syt1, eGFP-E-Syt1-ΔSMP, eGFP-E-Syt1-ΔDE and used in Fig. 6, F and G. Arrows indicate GFP signals for eGFP-empty vector and eGFP-E-Syt1 mutants transfected and E-Syt1 signals for E-Syt1 endogenous and eGFP-E-Syt1 mutants transfected. **(D)** Quantification of E-Syt1 expression levels normalized on ACTIN (loading control) in WT, DKO + GFP-E-Syt1 full length, DKO + GFP-E-Syt1-ΔSMP, and DKO + GFP-E-Syt1-ΔDE HeLa cells. The values plotted are the mean ± SEM from three biological replicates analyzed using one-way ANOVA, with Tukey's test for multiple comparisons. NS = not significant. Source data are available for this figure: SourceData FS5.

Video 1.    **The 3D EMCS in PERK proficient cells (FIB-SEM).** 3D video of the EMCS segmentation from a section of the FIB-SEM analysis in shCTR HeLa cell showing in red the mitochondria and in green the EMCS.

Video 2    . **PERK deficient cells display reduced 3D EMCS (FIB-SEM).** 3D video of the EMCS segmentation from a section of the FIB-SEM analysis in shPERK HeLa cell showing in red the mitochondria and in green the EMCS.

Video 3.    **From FIB-SEM images to EMCS segmentation in shCTR.** 3D FIB-SEM image series and animation showing reconstructed ER, mitochondria, and EMCS of shCTR HeLa cell. The mitochondria are labeled in red and ER in green.

Video 4.    **From FIB-SEM images to EMCS segmentation in shPERK.** 3D FIB-SEM image series and animation showing reconstructed ER, mitochondria and EMCS of shPERK HeLa cell. The mitochondria are labeled in red and ER in green.

