## [Peer Review File · The Journal of Cell Biology]

PERK recruits E-Syt1 at ER-mitochondria contacts for mitochondrial lipid transport and respiration

Maria Livia Sassano, Alexander van Vliet, Ellen Vervoort, Sofie Van Eygen, Chris Van den Haute, Benjamin Pavie, Joris Roels, Johannes V Swinnen, Marco Spinazzi, Leen Moens, Kristina Casteels, Isabelle Meyts, Paolo Pinton, Saverio Marchi, Leila Rochin, Francesca Giordano, Blanca Felipe Abrio, and Patrizia Agostinis

Corresponding Author(s): Patrizia Agostinis, KU Leuven and Blanca Felipe Abrio, KU Leuven

Review Timeline:

Submission Date:	2022-06-02
Editorial Decision:	2022-07-13
Revision Received:	2022-12-07
Editorial Decision:	2022-12-27
Revision Received:	2023-01-11

Monitoring Editor: William Prinz

Scientific Editor: Dan Simon

Transaction Report:

DOI: <https://doi.org/10.1083/jcb.202206008>

July 13, 2022

Re: JCB manuscript #202206008

Prof. Patrizia Agostinis
KU Leuven
Laboratory of Cell Death Research and Therapy
Herestraat 49 - O&N1bis - Bus 802
Leuven 3000
Belgium

Dear Prof. Agostinis,

Thank you for submitting your manuscript titled "PERK recruits E-Syt1 at ER-mitochondria contacts for mitochondrial lipid transport and respiration." The manuscript was assessed by expert reviewers, whose comments are appended to this letter. We invite you to submit a revision if you can address the reviewers' key concerns, as outlined here.

You will see that all three reviewers are enthusiastic about your study, an opinion we share. However, they also raise significant concerns. Chief among these is the request for firmer evidence that lipid transport to mitochondria is truly decreased after knockdown of PERK or E-Syt1 (Reviewer 3, pt. #1). Reviewers 2 and 3 also ask for more details about your lipid analysis of mitochondria and total cellular lipids levels (Reviewer 3, pt #2 and Reviewer 2, pt #1). Hopefully, these concerns can be addressed without additional experimentation, but if more is necessary it is not required to determine the levels of species in each lipid class, though this information is welcome if it is available. Reviewer 3 asks for confirmation that PERK and E-Syt1 interact using BiFC or FLIM-FRET and PERK overexpression (pts. #4 and 5). These results would be welcome but are not necessary for the revision. All the other concerns of Reviewers 1 and 2 and the minor concerns of Reviewer 3 should be addressed.

GENERAL GUIDELINES:

Text limits: Character count for an Article is < 40,000, not including spaces. Count includes title page, abstract, introduction, results, discussion, and acknowledgments. Count does not include materials and methods, figure legends, references, tables, or supplemental legends.

Figures: Articles may have up to 10 main text figures. Figures must be prepared according to the policies outlined in our Instructions to Authors, under Data Presentation, <https://jcb.rupress.org/site/misc/ifora.xhtml>. All figures in accepted manuscripts will be screened prior to publication.

*****IMPORTANT:** It is JCB policy that if requested, original data images must be made available. Failure to provide original images upon request will result in unavoidable delays in publication. Please ensure that you have access to all original microscopy and blot data images before submitting your revision. ***

Supplemental information: There are strict limits on the allowable amount of supplemental data. Articles may have up to 5 supplemental figures. Up to 10 supplemental videos or flash animations are allowed. A summary of all supplemental material should appear at the end of the Materials and methods section.

Please note that JCB now requires authors to submit Source Data used to generate figures containing gels and Western blots with all revised manuscripts. This Source Data consists of fully uncropped and unprocessed images for each gel/blot displayed in the main and supplemental figures. Since your paper includes cropped gel and/or blot images, please be sure to provide one Source Data file for each figure that contains gels and/or blots along with your revised manuscript files. File names for Source Data figures should be alphanumeric without any spaces or special characters (i.e., SourceDataF#, where F# refers to the associated main figure number or SourceDataFS# for those associated with Supplementary figures). The lanes of the gels/blots should be labeled as they are in the associated figure, the place where cropping was applied should be marked (with a box), and molecular weight/size standards should be labeled wherever possible. Source Data files will be made available to reviewers during evaluation of revised manuscripts and, if your paper is eventually published in JCB, the files will be directly linked to specific figures in the published article.

Source Data Figures should be provided as individual PDF files (one file per figure). Authors should endeavor to retain a minimum resolution of 300 dpi or pixels per inch. Please review our instructions for export from Photoshop, Illustrator, and PowerPoint here: <https://rupress.org/jcb/pages/submission-guidelines#revised>.

The typical timeframe for revisions is three to four months. While most universities and institutes have reopened labs and allowed researchers to begin working at nearly pre-pandemic levels, we at JCB realize that the lingering effects of the COVID-19 pandemic may still be impacting some aspects of your work, including the acquisition of equipment and reagents. Therefore, if you anticipate any difficulties in meeting this aforementioned revision time limit, please contact us and we can work with you to find an appropriate time frame for resubmission. Please note that papers are generally considered through only one revision cycle, so any revised manuscript will likely be either accepted or rejected.

Thank you for this interesting contribution to Journal of Cell Biology. You can contact us at the journal office with any questions, cellbio@rockefeller.edu or call (212) 327-8588.

Sincerely,

William Prinz, PhD
Monitoring Editor
Journal of Cell Biology

Dan Simon, PhD
Scientific Editor
Journal of Cell Biology

Reviewer #1 (Comments to the Authors (Required)):

Sassano et al. investigate the significance of the ER kinase PERK for ER-mitochondria lipid transfer. Using a variety of systems (HeLa, patient cells, ko, kd), they find that the amounts of PC, PI, and PE are reduced when PERK proteins are compromised. From previous MS results, they suspect that E-Syt1 could be responsible for an interaction with PERK and subsequent promotion of lipid transfer. This is analyzed with an elegant fluorescent lipid transfer assay that led to high quality data. The research is also connected to Wolcott-Rallison syndrome patient fibroblasts. The novelty of the research lies in the identification of E-Syt1 in a lipid transfer function for PERK. However, the role of PERK in this mechanism remains a little vague. Does it mediate E-Syt1 localization? Or does it promote a lipid transfer function? Some additional investigation could provide some information about this question. Overall, this remains a very interesting study that adds to the PERK function array, demonstrating yet again that this kinase is far from a simple unfunctional UPR sensor protein.

Specific points

1. The co-immunoprecipitation assays are not ideal, especially Figure 2A. At the moment, expression of wild type and mutant PERK are not even and the relative amounts cannot be assessed. The authors must achieve equal expression for both constructs by adjusting the plasmid amounts and provide the results on uncut gels, providing the relative amount of lysate that gives an idea how much protein was precipitated. This is important, since currently it looks like KD PERK binds E-Syt1 better than wild type. Also, a non-transfected control should be provided for the transfection.
2. A test for the specificity of the E-Syt1 antibody in immune-EM should be provided.
3. Why are the cytochrome c levels reduced upon PERK knockdown in Figure 2A? This could suggest a reduction or degradation of mitochondria.
4. In Figure 3G, it appears the delta DE construct shows Golgi overlap. This should be analyzed and quantified. This should be expanded into PERK wt and ko MEF cells to investigate if PERK interaction controls E-Syt1 retention in the ER.
5. If the defect indeed stems from lipid imbalance, then the supplementation of the growth medium with lyso-PE should rescue it. The authors should test this with their key assays.
6. The significance of the FIB data at the beginning of the paper is unclear, given it is later not explored in the context of mutants and their rescue.

Minor points:

1. Rather than calling the kinase-dead mutant "KD" in the figures, it would be preferable to call it by its mutation to avoid confusion with "knockdown".

Reviewer #2 (Comments to the Authors (Required)):

In this manuscript from Sassano et al, the authors describe a new role for E-Syt1 in delivering lipids from the ER to the mitochondria and further demonstrate that this role is supported by interaction with PERK which facilitates the formation and recruitment to ER-mito contact sites. A strength of the paper is that the majority of their results are demonstrated by multiple orthogonal approaches. Lipid transfer is inferred by the accumulation of exogenously added NBD-PS into mitochondria as well as by changes in the lipidomics of isolated mitochondria. PERK-ESyt-1 complexes are shown by pull-down with either PERK or ESyt1 overexpressed, ESyt-1 localization to MAM sites was established with both immunogold-EM and fluorescence colocalization of overexpressed tagged ESyt1. The authors also carry out domain analysis, determining that ESyt1 engages PERK via two of its C2 domains and that PERK activity in this pathway is independent of its kinase function. Consistent with a role in mitochondrial homeostasis, disruption of this interaction, or depletion of either protein diminishes respiration.

The manuscript was a pleasure to read. There are a number of small caveats to fully interpreting the main conclusions of the work (e.g., are ESyt1 and PERK in direct contact or part of a larger complex), but these concerns were raised directly by the authors in their discussion, and in any event, do not detract from the primary biological conclusions. I have only very small points/suggestions.

1. The authors show reductions in the abundance of mitochondrial PC, PI, and PE under various knockout or knockdown conditions, which they quantify as nmol lipid/mg protein and which they interpret as a reduction in lipid transport. They go to some lengths to argue that mitochondria are not grossly misshapen or reduced in mass (detailed in their supplemental data figs 1&2), though they do measure loss of respiration consistent with prior literature on PE-depleted mitochondria. The authors should show what, if any, impacts occur for other lipids in the mitochondria. Is there a compensation in the levels of other lipids? Or alternatively, are all lipids now present at reduced lipid:protein ratios, in which case it is less important to highlight these lipids which the authors suggest are "known to require the ER-mitochondria shuttle".
2. The average values in Fig 2E look surprisingly high, given the several individual points that were recorded as zero ESyt1 at either the MAM or the ER-PM. In fact, the average values appear to be the average of the non-zero cluster of points.
3. The controls throughout the manuscript are well done, why don't the authors also show the controls related to ER stress tests on figure 4?
4. The false-colored images to interpret the FIB-SEM were helpful, however the contact sites, depicted in white, are difficult to see. Another color choice might make this key point more obvious.

Reviewer #3 (Comments to the Authors (Required)):

In this work, Sassano and colleagues investigate the role of PERK-E-Syt1 complex in mitochondria lipid transport and function. E-Syt1 is well known to be located at ER-PM and to regulate ER-PM tethering and PM lipid homeostasis, both activities being regulated by Ca²⁺. Thus, the finding that E-Syt1 can also play a role at ER-mitochondria contact sites (EMCs) is really exciting. The authors show that the localization at EMCs is regulated by its interaction with PERK, a protein known to be located at EMCs and to regulate EMCs and some mitochondria function in addition to its role in ER UPR pathway. They identify the C2 domains DE of E-Syt1 as key domains for PERK interaction. Then, they show that both PERK and E-Syt1 are involved in mitochondrial lipid transfer and mitochondria respiration and that those functions relies on PERK-Syt1 interaction. The interaction between E-Syt1 and PERK as well as the unexpected role of E-Syt1 in mitochondria function are convincing. However, to my opinion, the localization of E-Syt1-PERK complex at EMCs and its role in mitochondrial lipid transport need more investigation. I'm aware that showing the direct role of a protein in lipid trafficking is not an easy task as there is no perfect methods to do so. But some other experiments can be performed to further support the role of E-Syt1 in mitochondrial lipid transport.

Major comments

- Role in lipid transfer:

- 1- Quantification of lipid transfer defect by in vivo NBD-PS mitochondria labelling: (Fig 1 E/F/G/H, Fig2L, 3I) : we cannot rule out the possibility that the decrease in mitotracker/NBD fluorescence co-localization observed after 30 min could be link to a decrease incorporation of the NBD-PS in the mutant cells. As example, in figure 1E, the total level of NBD signal in shPERK cells seems lower than in control cells. Thus, to take into consideration this hypothesis, the level of NBD signal co-localizing with MitoTracker should be normalized by the total level of NBD obtained in the whole cell. This is also true for mitotraker that enter active mitochondria, so if mitochondria function is altered, as it is shown in Fig4 for shPERK or DKO, the total signal of mitochondria in cells might be lower or some less functional mitochondria might not be labeled. As NBD-PS can be converted to NBP-PE in mitochondria, another relevant experiment to show the role of PERK and E-Syt1 in ER-mitochondria lipid transport is to analyze the conversion of PS to PE as they can be easily separated by TLC (10.1016/j.str.2020.04.006). Then, by calculating the ratio NBD-PE/(NBD-PS + NBD-PE) and showing this ratio decrease in shlines vs control line, the authors will bring a more convincing proof of the role of PERK and E-Syt1 in lipid trafficking.
- 2- Lipidomic analysis of shPERK and shE-Syt1 mitochondria: lipidomic is a powerful tool to analysis the whole lipidome of cells. Why the authors look only at some lipids putatively imported? What is the total the level of phospholipids/mg of mitochondria?

What about CL, PS, PG, PA, DAG? Is there modifications of lipid species (e.g. different acyl chain composition of lipids)? From the Mat and Met part, it seems that other lipids have been analyzed, the authors have to show the results for other lipids and comment those results to show that the decrease is specific to a subset of lipid classes and not a general decrease of the ratio phospholipids/mg of protein link to a general defect in mitochondria biogenesis. Also it might be useful to analysis the lipids on the whole cell extract to confirm that the decrease is specific to mitochondria and not linked to a perturbation of the general lipid homeostasis in the sh cells due to ER stress as example.

3- Fig3I/J: the author should perform also those experiments with the E-Syt1- Δ SMP as a negative control to show that indeed the lipid transfer domain of E-Syt1, in addition to its interaction with PERK, is required for lipid transport.

- Involvement of PERK in E-Syt1 localization to EMCs:

4- To my opinion, the authors never show that E-Syt1 and PERK interact at EMCs. The data provided are indirect as they show that GFP-E-Syt1 co-localization with MitoTracker decreases in shPERK. But in Fig1A, the authors show that in shPERK cells, EMCs are decreased. Thus, the decrease in co-localization of GFP-E-Syt1 with MitoTracker might just be linked to a decrease in EMCs numbers in those cells. To directly show E-Syt1 and PERK interact at EMCs, the authors should use technics to study in vivo protein-protein interaction, such as BiFC or FLIM-FRET, to show that indeed the signal given by the interaction between those two proteins are at the ER-mitochondria interface. Then this kind of technics can be use also to study the domains required for PERK interaction.

5- Using PERK overexpressor lines could also be a good strategy to show that PERK is required for the E-Syt1 localization at EMCs (by immunolabeling as example, to show that the % of E-Syt1 in MAM increase compared to WT).

Minor:

6- Figure 2A: Co-IPs results are not very convincing as also a high signal for E-Syt1 is observed in the negative control. The authors should probe the western with another antibody against a ER membrane protein not thought to interact with PERK.

7- I don't understand what is the difference between the lipid analyses performed in Fig2L and FigS2R on Syt1 KD. Why one is significantly different and not the other?

8- FigS3B/C: Why there are two bands for some constructions with E-Syt1 antibody but not for the GFP antibody? GFP signal is supposed to represent E-Syt1-GFP constructs level. Why there are some discrepancies between E-Syt1 and GFP antibodies signals (GFP-E-syt1 Δ CDE appears less expressed than others with E-Syt1 antibodies and more expressed with anti-GFP) ?

9- Cell fractionation experiments show that the level of E-Syt1 in the MAM decreases in PERK KD cells. Then they use co-localization analyses between GFP-Syt1 and MitoTracker to show that the level of E-syt1 co-localization decreases in the shPERK and that this is rescue when PERK is expressed (Fig 1J). Can the authors provide a control (western blot or quantification of total E-Syt1-GFP on cells) to show that the level of expression of E-Syt1-GFP is the same between the different lines?

Answer to the Reviewers

Reviewer #1

Sassano et al. investigate the significance of the ER kinase PERK for ER-mitochondria lipid transfer. Using a variety of systems (HeLa, patient cells, ko, kd), they find that the amounts of PC, PI, and PE are reduced when PERK proteins are compromised. From previous MS results, they suspect that E-Syt1 could be responsible for an interaction with PERK and subsequent promotion of lipid transfer. This is analyzed with an elegant fluorescent lipid transfer assay that led to high quality data. The research is also connected to Wolcott-Rallison syndrome patient fibroblasts. The novelty of the research lies in the identification of E-Syt1 in a lipid transfer function for PERK. However, the role of PERK in this mechanism remains a little vague. Does it mediate E-Syt1 localization? Or does it promote a lipid transfer function? Some additional investigation could provide some information about this question. Overall, this remains a very interesting study that adds to the PERK function array, demonstrating yet again that this kinase is far from a simple unifunctional UPR sensor protein.

General answer:

We thank the Reviewer 1 for the enthusiasm about our study and the precise summary of the main discoveries of our manuscript. Our data support a model whereby PERK contributes to ER-mitochondria lipid transferring indirectly, by facilitating E-Syt1 localization at the EMCS. This proposed model is supported by different complementary findings, which we highlight below:

- 1) Subcellular fractionation analysis of EMCS shows a reduced amount of E-Syt1 in PERK deficient cells (Fig. 3F-H; Fig. S2C-E).
- 2) Live imaging shows that while the fraction of E-Syt1 colocalizing with mitochondria in PERK deficient cells is significantly reduced in comparison to PERK proficient cells (shCTR), the expression of the catalytically inactive PERK, which retains its scaffolding function, rescues it (Fig. 3 J,K).
- 3) The expression of the E-Syt1 mutant that abolishes PERK-E-Syt1 interaction (GFP-E-Syt1 Δ DE) in E-Syt1/2 DKO HeLa cells shows a significantly reduced mitochondria colocalization compared to the full length E-Syt1 (Fig. 4 G,H).

Furthermore, as suggested by the Reviewer, to further assess whether PERK-E-Syt1 interaction regulates E-Syt1 retention in the ER, we performed complementary fluorescence studies accompanied by quantification, using STIM1 as additional tubular ER protein that has been reported to colocalize with E-Syt1 in resting and following ER-Ca²⁺ store depletion conditions ((1)), in combination with E-Syt1FL or the E-Syt1 Δ DE mutant, in PERK^{+/+} and PERK^{-/-} MEF and DKO HeLa cells.

In the revised manuscript, we now show that:

- I) the presence of E-Syt1 at the ER is enhanced in the absence of PERK in MEFs (new Fig. S4 J,K),
- II) when expressed in PERK^{+/+} MEFs and DKO HeLa cells, the E-Syt1 Δ DE mutant -which does not interact with PERK- is retained in the ER and this to a significantly greater extent than the E-Syt1 FL (new Fig. 4I,J; Fig. S4 J,K).
- III) in PERK^{-/-} cells both E-Syt1FL and the E-Syt1 Δ DE mutant show an increased ER localization (new Fig. S4J,K).
- IV) in DKO cells the expression of the E-Syt1 Δ SMP mutant (new Fig. 5E,F), which is required for E-Syt1 PL binding and trafficking (2) and can still bind PERK (Fig. 4C,E), fails to recover the mitochondrial PS defects caused by loss of E-Syt1 (new Fig. 5E,F).

These additional studies together with our previous E-Syt1/mitochondria colocalization results, portray that disruption of the PERK-E-Syt1 interaction or the absence of PERK, result in the delocalization of the EMCS-associated fraction of E-Syt1 to the ER. They also support our proposed model suggesting that PERK recruits a pool of E-Syt1 to the EMCS to regulate E-Syt1-mediated transfer of PLs to the mitochondria.

Answers to specific points

1. *The co-immunoprecipitation assays are not ideal, especially Figure 2A. At the moment, expression of wild type and mutant PERK are not even and the relative amounts cannot be assessed. The authors must achieve equal expression for both constructs by adjusting the plasmid amounts and provide the results on uncut gels, providing the relative amount of lysate that gives an idea how much protein was precipitated. This is important, since currently it looks like KD PERK binds E-Syt1 better than wild type. Also, a non-transfected control should be provided for the transfection.*

Answer: We thank the Reviewer 1 for the constructive comment and we agree that Figure 2A in particular was missing some relevant controls. We repeated a whole new set of co-IPs under conditions that would allow us to monitor and control comparable expression of the PERK full length/PERK kinase dead mutant and E-Syt1. To this end, we performed co-IPs using PERK-myc or the kinase dead mutant PERK^{K618A}-myc with E-Syt1-GFP.

A representative uncut WB from these co-IPs is shown in Fig. 3A. The western blot shows now the missing controls namely; a negative control of non-transfected cells, negative controls of 'GFP only' co-transfected with PERK-myc and the PERK^{K618A}-myc and an additional control of an unrelated integral ER and EMCS-associated protein, calnexin (CNX) (as also per request of Reviewer 3, minor point 6).

The new WB of Fig. 3A shows that i) immunoblot of total lysates shows that the expression of PERK-myc and the PERK^{K618A}-myc is similar; ii) no signal is detected in the untransfected cells,

ii) CNX is found in the total lysate but not in the co-IP; iii) co-transfection with GFP alone does not detect PERK and iv) both PERK FL and even more the PERK kinase dead are pulled down in cells expressing GFP-E-Syt1, confirming the same trend shown in our previous IPs, of an increased E-Syt1 association with PERK kinase dead (yet not statistically significant, when quantifying the biological repeats). As requested we quantified the relative amount of GFP-E-Syt1 immune-precipitated protein on total lysate abundance related to the WBs of Fig. 3A (Fig.S2B) and Fig. 4A,C (Fig.S4 D,F) were all E-Syt1 mutants were used.

Please note that in this set of co-IP experiments using co-transfection of myc and eGFP-tagged proteins we preferred to use a fluorescent image system (Chemidoc MP) to increase sensitivity, since the detection of the immunoreactive bands by chemoluminescence (ECL) required increased exposure times causing decreased signal to background noise ratio. While improving the sensitivity by using the fluorescence system, we can still detect just above the PERK band in the total lysates and further only in the co-IP conditions, the residual GFP-E-Syt1 signal as a spectral bleed-through. We explained this in the main text and, since PERK-myc and GFP-E-Syt1 display a close but still distinguishable molecular weight in PAGE-SDS, we indicated these bands with an arrow and by the molecular markers in the Western Blot.

We hope that the Reviewer will be satisfied with this new set of co-IPs which confirm our previous ones and as validated by other results shown in the manuscript especially when co-IPs are performed from the crude mitochondrial fractions.

2. A test for the specificity of the E-Syt1 antibody in immune-EM should be provided.

Answer: We apologize for the lack of specific information about the antibody used in our IEM studies of the previous version of our manuscript, and thank the Reviewer for raising this point and allowing us to properly correct this misunderstanding.

Here we have used an anti-eGFP to study the localization of eGFP-E-Syt1 in HeLa cells by IEM, as in multiple previous IEM studies including those on E-Syt1 (3) and on ORP5/8, lipid transfer proteins also localized at ER subdomains in contact with mitochondria (i.e. (4,5),) that confirm the specificity and suitability of this antibody and approach. We regret that this information was missing from the previous version of our manuscript, and we have now added it in the results text and in the figure legend and method section.

3. Why are the cytochrome c levels reduced upon PERK knockdown in Figure 2A? This could suggest a reduction or degradation of mitochondria.

Answer: We thank the Reviewer 1 for pinpointing this inconsistency. We do not think that discrepancies in the levels of cytochrome c which are detected *only in the total lysate fraction* reflect alteration in levels of cytochrome c caused by an increased degradation or reduced amount of mitochondria in the PERK silenced or KO cells under steady state conditions. This argument is supported by different observations: i) the total amount of cytochrome c of the isolated mitochondria fraction (obtained by subcellular fractionation after homogenization) is equal in control or PERK silenced HeLa cells (Fig. S1F) or PERK ^{+/+} and PERK ^{-/-} MEFs (Fig. S1G);

ii) the levels of the outer mitochondrial protein VDAC, from total lysate fractions do not vary;
iii) additional FACS data (new Fig. S1D) show that mitochondrial mass as measured by Mitotracker green is not significantly different in PERK proficient or deficient cells. Also, the discrepancy found in the detection of cytochrome c in the *total lysate fraction* are not significant, as shown in Rebuttal Letter Fig. 1A (**RL** Fig. 1A) when quantifying the western blots from Fig. 3F and Fig. S1F. Based on these observations we think that some discrepancy is possibly generated by a differential solubilization of proteins of the outer (e.g. VDAC, whose presence indeed in the total lysate fractions does not change) and inner membrane (cytochrome c) of mitochondria during the preparation of the total lysates (see also (6)), which is instead minimized by the homogenization during the preparation of the “mito pure”. However, after excluding the possibility that the amount of mitochondria is different between conditions, we would consider the “mito pure” fraction as more precisely informative in terms of mitochondrial abundances. A note has been added in the Materials and Methods.

4. In Figure 3G, it appears the delta DE construct shows Golgi overlap. This should be analyzed and quantified. This should be expanded into PERK wt and ko MEF cells to investigate if PERK interaction controls E-Syt1 retention in the ER.

Answer: We thank the Reviewer for these comments. As suggested, we performed labeling of the Golgi with CellLight™ Golgi-RFP, BacMam 2.0 in cells expressing the PERK-interacting defective E-Syt1 Δ DE mutant and full length (FL) E-Syt1, both in DKO HeLa cells and in PERK^{+/+} and PERK^{-/-} MEFs. As shown in the Rebuttal Figure 1B-E (**RL** Fig. 1B-E) both E-Syt1 FL and its E-Syt1 Δ DE mutant partially and to the same extent colocalize with the Golgi of PERK^{+/+} or PERK^{-/-} MEFs and DKO cells, suggesting that PERK presence or PERK-E-Syt1 interaction are not influential in this regard. However, in order not to overload the MS with additional colocalization studies, we prefer to show these results to the Reviewer in this Rebuttal Letter.

Moreover, as detailed above in the answer of the general comments, we performed additional colocalization/rescue experiments with full length E-Syt1 and the E-Syt1 Δ DE mutant and STIM1 both in PERK^{+/+} and PERK^{-/-} MEFs and DKO HeLa cells, strengthening the concept that the presence of PERK and its interaction with E-Syt1 regulate the relocation of a pool of E-Syt1 to the EMCS, as a mechanism that favors PL transport (new Fig.4I-J; Fig. S4J,K).

5. If the defect indeed stems from lipid imbalance, then the supplementation of the growth medium with lyso-PE should rescue it. The authors should test this with their key assays.

Answer: We thank the reviewer for suggesting this approach. We agree that the supplementation of lyso-PE could be a strategy to rescue the deficiency of PE in the mitochondria and thus the phenotype observed in cells with perturbed PERK-E-Syt1 interaction. However, the rescue effects of lyso-PE would occur only if i) lyso-PE directly localizes at the mitochondria, bypassing the need to be transported via the EMCS in PERK silenced cells, and ii) independently of PERK-E-Syt1 interaction.

Literature studies however reported that exogenous lyso-PE is first translocated across the PM and then reaches the EMCS through a pathway that is still elusive (7). At the EMCS lyso-PE is then acylated by Ale1p, an acyl-CoA-dependent acyltransferase enriched at the EMCS (8,9). The PE made by this pathway can then reach the mitochondria (10,11). Also this alternative pathway of production of PE has a limited capacity to replace PE made in the mitochondria (12). A further complication we encountered was that the only commercially available option of lyso-PE NBD bound (to study its cellular trafficking) with the same carbon number as 18:1-06:0 NBD PS, which we used in the manuscript (<https://avantilipids.com/product/810194>), was lyso-PE 18:1 (<https://avantilipids.com/product/810127%E2%80%8B>). While 18:1-06:0 NBD PS presents the NBD group bound to the tail of the PL, the commercially available lyso-PE 18:1 presents the NBD group bound to the head PL group, which might affect the physicochemical characteristic of this PL. Unfortunately, 6:0 NBD lyso PE is not commercially available and to customize the synthesis of 6:0 NBD lyso PE or 18/1 lyso PE (which was extremely costly) was unfortunately beyond the time allowed for the revision of this paper.

We then considered as a possible alternative approach the direct addition of 18:1-06:0 NBD-PE into the medium. This PL is NBD tail bound, contains the same number of C and unsaturation in the acyl chain. When we then performed live imaging to visualize its sub-cellular localization, we found that after 30 min only about 20% of the 18:1-06:0 NBD PE localized at the mitochondria, while most of the fluorescent PE signal showed a diffuse cytosolic/ER redistribution -as shown in the Rebuttal Letter Fig. 2A,B (RL Fig. 2 A,B)- and expanding the time of incubation, resulted in an inconsistent pattern of mitochondrial localization within the cell's population (not shown). Under these conditions, we could not detect differences in lower mitochondrial respiration of the PERK deficient cells as shown in the Rebuttal Letter Fig. 2C,D (RL Fig.2 C,D). Hence these results remain of dubious interpretation since the exogenous addition of PE resulted in only a limited and inconsistent localization at the mitochondria, an effect that impair the interpretation of cell population assays, such as OCR with the Seahorse analyzer. Also altering abundance of PE at the ER, can affect other pathways as LD synthesis (13) or ER stress (14,15), adding confounding factors.

Altogether, while we tried our best to appropriately use the approach suggested by the Reviewer 1, as shown by the results in our RL Figure 2 A-D, the lack of precise information about the trafficking mechanisms of exogenously added and fluorescently labelled PE, its main retention in the cytosol/ER and inconsistent pattern of mitochondria redistribution among cells, do not allow us to draw main conclusions based on these assays.

6. *The significance of the FIB data at the beginning of the paper is unclear, given it is later not explored in the context of mutants and their rescue.*

Answer: We thank the reviewer for pointing this out. Our previous study (16) showed the functional role of PERK in the regulation of ER-mitochondria contacts in a model of ER stress. In that study we used colocalization studies to image ER-mitochondria appositions in MEFs by confocal microscopy. Hence, here we considered of critical importance to first validate the

effects of PERK silencing on ER-mitochondria appositions in a different cellular model (HeLa cells), under homeostatic conditions and in particular, given the limit of the Z resolution of confocal microscopy, using FIB-SEM, in order to obtain 3D images with superior Z-axis resolution. We regret if this was not sufficiently clear and have specified better this point at the beginning of the Result section. We hope that the Reviewer appreciates our effort to use a variety of state of the art techniques in the field of contact sites, including but not limited to FIB-SEM and IEM, to fully explore and validate the functional effects of PERK-E-Syt1 axis unraveled by our studies, while considering that the effort and operational costs of FIB-SEM analysis prevented us to use this sophisticated approach as a read out of functional studies.

Minor points:

1. *Rather than calling the kinase-dead mutant "KD" in the figures, it would be preferable to call it by its mutation to avoid confusion with "knockdown".*

Answer: We thank the reviewer and we changed it accordingly through the text and Figures.

Reviewer #2

In this manuscript from Sassano et al, the authors describe a new role for E-Syt1 in delivering lipids from the ER to the mitochondria and further demonstrate that this role is supported by interaction with PERK which facilitates the formation and recruitment to ER-mito contact sites. A strength of the paper is that the majority of their results are demonstrated by multiple orthogonal approaches. Lipid transfer is inferred by the accumulation of exogenously added NBD-PS into mitochondria as well as by changes in the lipidomics of isolated mitochondria. PERK-E-Syt-1 complexes are shown by pull-down with either PERK or E-Syt1 overexpressed, E-Syt-1 localization to MAM sites was established with both immunogold-EM and fluorescence colocalization of overexpressed tagged E-Syt1. The authors also carry out domain analysis, determining that E-Syt1 engages PERK via two of its C2 domains and that PERK activity in this pathway is independent of its kinase function. Consistent with a role in mitochondrial homeostasis, disruption of this interaction, or depletion of either protein diminishes respiration.

The manuscript was a pleasure to read. There are a number of small caveats to fully interpreting the main conclusions of the work (e.g., are E-Syt1 and PERK in direct contact or part of a larger complex), but these concerns were raised directly by the authors in their discussion, and in any event, do not detract from the primary biological conclusions. I have only very small points/suggestions.

General Answer: We thank the Reviewer 2 for the appraisal of our study.

1. The authors show reductions in the abundance of mitochondrial PC, PI, and PE under various knockout or knockdown conditions, which they quantify as nmol lipid/mg protein and which they interpret as a reduction in lipid transport. They go to some lengths to argue that mitochondria are not grossly misshapen or reduced in mass (detailed in their supplemental data figs 1&2), though they do measure loss of respiration consistent with prior literature on PE-depleted mitochondria. The authors should show what, if any, impacts occur for other lipids in the mitochondria. Is there a compensation in the levels of other lipids? Or alternatively, are all lipids now present at reduced lipid:protein ratios, in which case it is less important to highlight these lipids which the authors suggest are "known to require the ER-mitochondria shuttle".

Answer: We thank the Reviewer for this constructive suggestion. As suggested by the Reviewer we performed additional lipidomics studies from total cellular fractions and isolated mitochondria of shCTR cells, shPERK or shE-Syt1 HeLa cells in order to support the conclusion that the observed defects in PC, PE and PI reflect mitochondria perturbations, which are functionally linked to defects in respiratory capacity, and not a general lipid dyshomeostasis. We also performed additional lipidomics in shCTR cells, shPERK or shE-Syt1 HeLa cells.

Expanding these lipidomics analysis, which are now integrated in the new Fig. 2C-F, Fig. 3L, as biological replicates and further in Fig.S1H-J, Fig.S3B-E, we show that:

- 1) Overall cellular abundances of PC, PI, PE and PG are not significantly altered by loss of PERK or E-Syt1 (new Fig. S1J; Fig.S3C), while we confirm their reductions in the purified mitochondrial fractions (new Fig. 2C for shCTR and PERK silenced HeLa cells and Fig.2D for WT and PERK-deficient MEFs, and Fig.3L for shCRT and E-Syt1 silenced HeLa cells). We also included in the manuscript (new Fig. S1J for PERK; Fig.S3C for E-Syt1) the overall cellular abundances of PS as we considered important to show that its total levels were unchanged, since PS is the precursor of mitochondrial PE.
- 2) Total level of phospholipids/mg of mitochondria is not altered by the downregulation or KO of PERK (new Fig. 2E,F) or E-Syt1 silencing (new Fig. S3D), nor is their mitochondrial mass (Mitotracker green staining Fig. S1D; Fig. S3G).
- 3) A trend in an increased mitochondrial DG content (Fig. S1I; Fig. S3E) is observed, in both PERK and E-Syt1 silenced cells suggesting the possibility of a compensation in the levels of other lipids.
- 4) We observed a reduced level of mitochondrial PG (Fig. S1H; Fig.S3B). However it should be noted that the mitochondrial content of PG which serves as a major precursor of CL (17,18), is quite low. Unfortunately PA and CL could not be detected due to poor peak shapes with the HILIC method used. PA is a highly acidic lipid class which causes very broad and tailing chromatographic peaks on the standard HILIC method that we use for lipidomics (which also seriously reduces the sensitivity). Besides poor peak shapes, for CL we also observed unpredictable retention times for different putative species so we erred on the side of caution and do not report CL species with this method.

In conclusion, these additional experiments further suggest that major PL classes of mitochondrial PLs -and in particular those known to be trafficked through the EMCS- are similarly reduced by the silencing of PERK or E-Syt1, while the overall mitochondrial lipid:protein ratio is not changed. They also highlight that these defects are specific to mitochondria and not caused by perturbation of the general lipid homeostasis.

2. The average values in Fig 2E look surprisingly high, given the several individual points that were recorded as zero ESyt1 at either the MAM or the ER-PM. In fact, the average values appear to be the average of the non-zero cluster of points.

Answer: We thank the reviewer for highlighting this discrepancy. The raw data were correct and we included as well the “zero” values but the Y axis of the graph was cut therefore hiding some data points. The Y axis has been adjusted in Fig. 3E.

3. The controls throughout the manuscript are well done, why don't the authors also show the controls related to ER stress tests on figure 4?

Answer: We thank the Reviewer for this suggestion and moved the ER stress control in the new Fig. 6E.

4. The false-colored images to interpret the FIB-SEM were helpful, however the contact sites, depicted in white, are difficult to see. Another color choice might make this key point more obvious.

Answer: We thank the reviewer and we changed accordingly the color, showing the contact sites in light blue in the revised version of the manuscript.

Reviewer #3

In this work, Sassano and colleagues investigate the role of PERK-E-Syt1 complex in mitochondria lipid transport and function. E-Syt1 is well known to be located at ER-PM and to regulate ER-PM tethering and PM lipid homeostasis, both activities being regulated by Ca²⁺. Thus, the finding that E-Syt1 can also play a role at ER-mitochondria contact sites (EMCs) is really exciting. The authors show that the localization at EMCs is regulated by its interaction with PERK, a protein known to be located at EMCs and to regulate EMCs and some mitochondria function in addition to its role in ER UPR pathway. They identify the C2 domains DE of E-Syt1 as key domains for PERK interaction. Then, they show that both PERK and E-Syt1 are involved in mitochondrial lipid transfer and mitochondria respiration and that those functions relies on PERK-Syt1 interaction.

The interaction between E-Syt1 and PERK as well as the unexpected role of E-Syt1 in mitochondria function are convincing. However, to my opinion, the localization of E-Syt1-PERK complex at EMCs and its role in mitochondrial lipid transport need more investigation. I'm aware that showing the direct role of a protein in lipid trafficking is not an easy task as there is no perfect methods to do so. But some other experiments can be performed to further support the role of E-Syt1 in mitochondrial lipid transport.

General Answer: We thank the Reviewer for the enthusiasm in assessing our findings.

Major comments

- Role in lipid transfer:

1- Quantification of lipid transfer defect by in vivo NBD-PS mitochondria labelling: (Fig 1 E/F/G/H, Fig2L, 3I) : we cannot rule out the possibility that the decrease in mitotracker/NBD fluorescence co-localization observed after 30 min could be link to a decrease incorporation of the NBD-PS in the mutant cells. As example, in figure 1E, the total level of NBD signal in shPERK cells seems lower than in control cells. Thus, to take into consideration this hypothesis, the level of NBD signal co-localizing with MitoTracker should be normalized by the total level of NBD obtained in the whole cell. This is also true for mitotraker that enter active mitochondria, so if mitochondria function is altered, as it is shown in Fig4 for shPERK or DKO,

the total signal of mitochondria in cells might be lower or some less functional mitochondria might not be labeled. As NBD-PS can be converted to NBD-PE in mitochondria, another relevant experiment to show the role of PERK and E-Syt1 in ER-mitochondria lipid transport is to analyze the conversion of PS to PE as they can be easily separated by TLC (10.1016/j.str.2020.04.006). Then, by calculating the ratio $NBD-PE/(NBD-PS + NBD-PE)$ and showing this ratio decrease in shlines vs control line, the authors will bring a more convincing proof of the role of PERK and E-Syt1 in lipid trafficking.

Answer: We thank the reviewer for these constructive comments and suggestions.

We agree that in the representative picture of Fig.1E the signal of NBD-PE looks a bit lower and might suggest a reduced cellular incorporation of the fluorescently labelled PL. This 'effect' might be due to the fact that while in shCTR NBD-PS enters mitochondria where it is rapidly converted into NBD-PE in the shPERK cells, the fluorescent signal of the NBD-PS remains diffuse in the cytosol/ER resulting in a less bright/more diffuse staining. However, the colocalization assay provides a measure of the *ratio between the NBD-PS signal colocalizing with, in this case, the mitochondria, over the total NBD-PS signal* (19) and therefore the signal of NBD-PS is already normalized on the total fluorescent signal. This rules out the possibility that the observed defects in PS trafficking under conditions disrupting the PERK-E-Syt1 interaction are due to a reduced PS cellular entry.

Regarding the possibility that the total signal of mitochondria is lower in PERK or E-Syt1 deficient cells, which have defects in OCR (new Fig 6), we provide evidence that using the mitochondrial potential-independent Mitotracker green (Fig. S1D, Fig. S3G) no differences in mitochondrial mass in PERK or E-Syt1 silenced cells compared to their shCTR counterparts respectively, is detected. However, to further confirm this point we performed additional measurements of the mitochondrial membrane potential in shCTR and shPERK or shE-Syt1 cells. We expressed the TMRM/Mitotracker green ratio, Rebuttal Letter Fig 3A,B (RL Fig. 3A,B), which showed that mitochondrial potential is not significantly altered by either PERK or E-Syt1 silencing, supporting the results obtained with the Mitotracker deep red used in Fig.1 E-H, new Fig. 5C-F. It should be noted that in previous reports a decreased transport of PE to the mitochondria resulted in a reduced respiratory capacity, without reduction in the mitochondrial membrane potential, which instead, was reported to be slightly increased in absence of phosphatidylserine decarboxylase (PSD) (12).

As requested by the Reviewer, to confirm the role of PERK and E-Syt1 in ER-mitochondria lipid transport, we performed additional assays to evaluate the conversion of PS to PE by TLC separation. We compared shCTR with shPERK cells and E-Syt1/2 DKO cells with DKO cells repressing E-Syt1. Consistent with other observations (live imaging, lipidomics), we show that compared to their respective controls/counterparts, both shPERK (new Fig. 2A,B) and DKO cells (new Fig. 5A,B) show an impaired ability to convert NBD-PS to NBD-PE, and the transient rescue of E-Syt1 in DKO partially rescues this process in DKO cells (new Fig.5A,B). The quantification shows the ratios of $NBD-PE/(NBD-PS + NBD-PE)$ relative to their controls.

In conclusion, the additional controls and experiments suggested by the Reviewer 3, further support the relevance of the PERK-E-Syt1 axis in ER-mitochondrial lipid transport.

2- Lipidomic analysis of shPERK and shE-Syt1 mitochondria: lipidomic is a powerful tool to analysis the whole lipidome of cells. Why the authors look only at some lipids putatively imported? What is the total the level of phospholipids/mg of mitochondria? What about CL, PS, PG, PA, DAG? Is there modifications of lipid species (e.g. different acyl chain composition of lipids)? From the Mat and Met part, it seems that other lipids have been analyzed, the authors have to show the results for other lipids and comment those results to show that the decrease is specific to a subset of lipid classes and not a general decrease of the ratio phospholipids/mg of protein link to a general defect in mitochondria biogenesis. Also it might be useful to analysis the lipids on the whole cell extract to confirm that the decrease is specific to mitochondria and not linked to a perturbation of the general lipid homeostasis in the sh cells due to ER stress as example.

Answer: We thank the Reviewer for raising these interesting and constructive conjectures. As requested by the Reviewer we performed additional lipidomics studies from total cellular fractions and isolated mitochondria of shCTR cells, shPERK or shE-Syt1 HeLa cells in order to support the conclusion that the observed defects in PC, PE and PI reflect mitochondria perturbations, which are functionally linked to defects in respiratory capacity, and not a general lipid dyshomeostasis. We also performed additional lipidomics in of shCTR cells, shPERK or shE-Syt1 HeLa cells.

Expanding these lipidomics analysis, which are now integrated in the new Fig. 2C-F, Fig. 3L, as biological replicates and further in Fig.S1H-J, Fig.S3B-E, we show that:

- 1) Overall cellular abundances of PC, PI, PE and PG are not significantly altered by loss of PERK or E-Syt1 (new Fig. S1J; Fig.S3C), while we confirm their reductions in the purified mitochondrial fractions (new Fig. 2C for shCTR and PERK silenced HeLa cells and Fig. 2D for WT and PERK-deficient MEFs, and Fig. 3L for shCTR and E-Syt1 silenced Hela cells). We also included in the manuscript (new Fig. S1J for PERK; Fig. S3C for E-Syt1) the overall cellular abundances of PS as we considered important to show that its total levels were unchanged, since PS is the precursor of mitochondrial PE.
- 2) Total level of phospholipids/mg of mitochondria is not altered by the downregulation or KO of PERK (new Fig. 2E,F) or E-Syt1 silencing (new Fig. S3D), nor is their mitochondrial mass (Mitotracker green staining Fig. S1D; Fig. S3G).
- 3) A trend in an increased mitochondrial DG content (Fig. S1I; Fig. S3E) is observed, in both PERK and E-Syt1 silenced cells suggesting the possibility of a compensation in the levels of other lipids.
- 4) We observed a reduced level of mitochondrial PG (Fig. S1H; Fig.S3B). However it should be noted that the mitochondrial content of PG which serves as a major precursor of CL (17,18), is quite low. Unfortunately PA and CL could not be detected due to poor peak shapes with the HILIC method used. PA is a highly acidic lipid class which causes very

broad and tailing chromatographic peaks on the standard HILIC method that we use for lipidomics (which also seriously reduces the sensitivity). Besides poor peak shapes, for CL we also observed unpredictable retention times for different putative species so we erred on the side of caution and do not report CL species with this method.

- 5) There are no major changes in the acyl chains (e.g. number of carbons) of the major mitochondrial PLs investigated as shown in the Rebuttal Letter Fig. 3C (RL Fig. 3C).

In conclusion, these additional experiments further suggest that major PL classes of mitochondrial PLs -and in particular those known to be trafficked through the EMCS- are similarly reduced by the silencing of PERK or E-Syt1, while the overall mitochondrial lipid:protein ratio is not changed. They also highlight that these defects are specific to mitochondria and not caused by perturbation of the general lipid homeostasis.

3- Fig3I/J: *the author should perform also those experiments with the E-Syt1-ΔSMP as a negative control to show that indeed the lipid transfer domain of E-Syt1, in addition to its interaction with PERK, is required for lipid transport.*

Answer: We thank the reviewer for this suggestion. We performed as requested by the Reviewer additional NDB-PS imaging data in live DKO cells expressing the mCherry E-Syt1-ΔSMP mutant, which can still bind PERK but is unable to bind and transport PLs. These additional data integrated in new Fig. 5 E,F show that the expression of the mCherry E-Syt1-ΔSMP mutant fails to rescue the defects in NDB-PS mitochondrial localization. They provide further support to our model by showing that the lipid transfer domain of E-Syt1, in addition to its interaction with PERK, is required for lipid transport.

- Involvement of PERK in E-Syt1 localization to EMCs:

4- *To my opinion, the authors never show that E-Syt1 and PERK interact at EMCs. The data provided are indirect as they show that GFP-E-Syt1 co-localization with MitoTracker decreases in shPERK. But in Fig1A, the authors show that in shPERK cells, EMCs are decreased. Thus, the decrease in co-localization of GFP-E-Syt1 with MitoTracker might just be linked to a decrease in EMCs numbers in those cells. To directly show E-Syt1 and PERK interact at EMCs, the authors should use technics to study in vivo protein-protein interaction, such as BiFC or FLIM-FRET, to show that indeed the signal given by the interaction between those two proteins are at the ER-mitochondria interface. Then this kind of technics can be use also to study the domains required for PERK interaction.*

Answer: We thank the reviewer for raising this point and we agree that in vivo protein-protein interaction, with BiFC or FLIM-FRET, are more direct assays, which could be explored in future studies as their implementation here would require a significant extension of the time allocated for the revision.

However, we would like to point out that although we agree with the Reviewer that co-immunoprecipitations do not portray per se a direct interaction we respectfully disagree with the conclusion that we provide no evidence of the PERK-Esyt1 interaction at EMCS. In line with

this, we performed co-IP assays showing the interaction between PERK and E-Syt1 and E-Syt1 mutants from total lysate as well as from the "crude mitochondrial" fraction, which is enriched in the mitochondria and its surrounding membranes (Fig. 4C-F)(20). Moreover, this result was supported by cellular fractionation experiments where we isolated the EMCS and we quantify E-Syt1 abundance, normalized to the amount of total EMCS (specifically to the ratio of CNX; Fig. 3F-H; Fig.S2C-E).

Our live imaging data show that in PERK proficient, E-Syt1/2 DKO cells after re-expression of the E-Syt1 full length (FL) or its E-Syt1 mutants, E-Syt1 colocalizes with the mitochondria in a PERK-interaction dependent mechanism. The E-Syt1 PERK interacting deficient mutant (E-Syt1 Δ DE mutant) showed a significant reduction in the colocalization with the mitochondrial network in comparison to the E-Syt1FL protein. Considering that in our system E-Syt1 does not per se modulate the integrity of EMCS (Fig. S2J,K), we believe that PERK-mediated E-Syt1 relocation at ER-mitochondria apposition is independent of the number of EMCS.

In conclusion, we posit that the data presented in our MS support an important role of PERK in scaffolding E-Syt1 at the EMCS via their interaction.

5- Using PERK overexpressor lines could also be a good strategy to show that PERK is required for the E-Syt1 localization at EMCs (by immunolabeling as example, to show that the % of E-Syt1 in MAM increase compared to WT).

Answer: We thank the Reviewer for this suggestion. Unfortunately, overexpression of PERK in our hands often elicits its dimerization which causes loss of viability by activating the ATF4-CHOP pathway and/or remodeling of the actin cytoskeleton (when using the PERK kinase dead mutant, see (21)). Since both OE strategies are difficult to control in term of off-targets/unwanted effects, we did not pursue them.

Instead, in order to validate the specific involvement of PERK and its functional relevance as a scaffold for the E-Syt1 recruitment at the EMCS, through the study, beyond PERK silencing, we performed re-expression experiment of the KD mutant in PERK silenced cells (Fig3. A,J,K; Fig. 6A,B).

Minor:

6- Figure 2A: Co-IPs results are not very convincing as also a high signal for E-Syt1 is observed in the negative control. The authors should probe the western with another antibody against a ER membrane protein not thought to interact with PERK.

Answer: We thank the Reviewer 3 for the constructive comment and we agree that Figure 2A in particular was missing some relevant controls. Also considering the Reviewer 1 suggestions we repeated a whole new set of co-IPs under conditions that would allow us to monitor and control comparable expression of the PERK full length/PERK kinase dead mutant and E-Syt1. To

this end, we performed co-IPs using PERK-myc or the kinase dead mutant PERK^{K618A}-myc with E-Syt1-GFP.

A representative uncut WB from these co-IPs is shown in Fig. 3A. The western blot shows now the missing controls namely; a negative control of non-transfected cells, negative controls of 'GFP only' co-transfected with PERK-myc and the PERK^{K618A}-myc and an additional control of an unrelated integral ER and EMCS-associated protein, calnexin (CNX).

The new WB of Fig. 3A shows that i) immunoblot of total lysates shows that the expression of PERK-myc and the PERK^{K618A}-myc is similar; ii) no signal is detected in the untransfected cells, ii) CNX is found in the total lysate but not in the co-IP; iii) co-transfection with GFP alone does not detect PERK and iv) both PERK FL and even more the PERK kinase dead are pulled down in cells expressing GFP-E-Syt1, confirming the same trend shown in our previous IPs, of an increased E-Syt1 association with PERK kinase dead (yet not statistically significant, when quantifying the biological repeats). As requested we quantified the relative amount of GFP-E-Syt1 immune-precipitated protein on total lysate abundance related to the WBs of Fig. 3A (Fig.S2B) and Fig. 4A,C (Fig.S4 D,F) were all E-Syt1 mutants were used.

Please note that in this set of co-IP experiments using co-transfection of myc and eGFP-tagged proteins we preferred to use a fluorescent image system (Chemidoc MP) to increase sensitivity, since the detection of the immunoreactive bands by chemoluminescence (ECL) required increased exposure times causing decreased signal to background noise ratio. While improving the sensitivity by using the fluorescence system, we can still detect just above the PERK band in the total lysates and further only in the co-IP conditions, the residual GFP-E-Syt1 signal as a spectral bleed-through. We explained this in the main text and, since PERK-myc and GFP-E-Syt1 display a close but still distinguishable molecular weight in PAGE-SDS, we indicated these bands with an arrow and by the molecular markers in the Western Blot.

We hope that the Reviewer will be satisfied with this new set of co-IPs which confirm our previous ones and as validated by other results shown in the manuscript especially when co-IPs are performed from the crude mitochondrial fractions.

7- I don't understand what is the difference between the lipid analyses performed in Fig2L and FigS2R on Syt1 KD. Why one is significantly different and not the other?

Answer: In the main figures of the original manuscript, we reported technical replicates, and show in the supplementary figures the three biological replicates since the latter were performed using a different lipidomics platform and could not be pooled. In the revised version of the MS (also as per the request of Reviewer 2) we performed additional quantitative lipidomics analysis comparing overall cellular PL species with specific changes in the lipidome of isolated mitochondria, as described previously in point 2. Hence now we are able to pull independent biological experiments (Fig. 2C-F, Fig. 3L, Fig.S1H-J, Fig.S3B-E) done on the same lipidomics platform.

8- FigS3B/C: *Why there are two bands for some constructions with E-Syt1 antibody but not for the GFP antibody? GFP signal is supposed to represent E-Syt1-GFP constructs level. Why there are some discrepancies between E-Syt1 and GFP antibodies signals (GFP-E-syt1 Δ CDE appears less expressed than others with E-Syt1 antibodies and more expressed with anti-GFP)?*

Answer: we thank the reviewer for this clarification. The commercially available anti-E-Syt1 antibody, in our hand often detects a lower molecular weight band which is visible in several western blots and is possibly due to a nonspecific immunoreaction. We believe that a discrepancy between the anti-GFP and anti-E-Syt1 in detecting GFP-E-syt1 Δ CDE, could be due to the fact that while GFP is bound to the N-terminal of the protein, and thus the anti-GFP antibody is able to recognize all the mutants, the commercially available anti-E-Syt1 antibody is generated against the first 1-800 aa of the protein (data not shown in the antibody datasheet). The shortest mutant GFP-E-Syt1 Δ CDE, 1-633 aa, may thus be less robustly detected by the anti-E-Syt1 antibody because the region 634-800aa of the antigenic epitope is missing.

9- Cell fractionation experiments show that the level of E-Syt1 in the MAM decreases in PERK KD cells. Then they use co-localization analyses between GFP-Syt1 and MitoTracker to show that the level of E-syt1 co-localization decreases in the shPERK and that this is rescue when PERK is expressed (Fig 1J). Can the authors provide a control (western blot or quantification of total E-Syt1-GFP on cells) to show that the level of expression of E-Syt1-GFP is the same between the different lines?

Answer: We thank the reviewer for highlighting this. We provide below a WB showing that GFP-E-Syt1 is expressed to the same level in shCTR, shPERK and shPERK+PERK kinase dead. Because of the lack of space in the Figure S1 we opted to show these additional controls to the Reviewer only in the Rebuttal Letter Fig. 3D,E (**RL** Fig. 3 D,E), but we would be happy to include them in the Figure S1 if the Reviewer would find it relevant.

RL Figures and legends:

Figure 1

Figure 1.

- (A) CYTC abundance in shCTR and shPERK HeLa cells from total lysate (normalized on CNX). Mean \pm SEM, N=3 biological replicates analyzed using one sample t test.
- (B) Representative images of GFP-E-Syt1 full length or GFP-E-Syt1- Δ DE transiently transfected and co-stained with Golgi (Cell Light Golgi-RFP) in HeLa DKO cells. Scale bar 10 μ m.
- (C) Colocalization analysis of GFP E-Syt1 or GFP E-Syt1 Δ DE and Golgi in DKO HeLa cells (Manders M1 coefficient). Mean \pm SEM, N=3 biological replicates analyzed using unpaired Student's t test.
- (D) Representative images of GFP-E-Syt1 full length or GFP-E-Syt1- Δ DE transiently transfected and co-stained with Golgi (Cell Light Golgi-RFP) in PERK+/+ and PERK-/- MEFs cells. Scale bar 10 μ m.
- (E) Colocalization analysis of GFP E-Syt1 or GFP E-Syt1 Δ DE and Golgi in PERK+/+ and PERK-/- MEFs cells (Manders M1 coefficient). Mean \pm SEM, N=3 biological replicates analyzed using one-way ANOVA, with Tukey's test for multiple comparisons.

Figure 2

Figure 2.

(A) Representative images of NBD-PE co-stained with Mitotracker Far Red in shCTRL and shPERK HeLa cells. Scale bar 10 μm .

(B) Colocalization analysis of NBD-PE and Mitotracker Far Red in shCTRL and shPERK HeLa cells (Manders M1 coefficient). Mean \pm SEM, N=2 biological replicates.

(C,D) Oxygen consumption rate (OCR) of shCTRL, shPERK and shPERK+ NBD-PE HeLa cells in galactose media (C); quantification of basal respiration, ATP production and maximal respiration (D). Mean \pm SEM, N=2 biological replicates.

Figure 3

Figure 3.

(A) Mitochondrial membrane potential measured as TMRM geometrical mean intensity (MFI) normalized on Mitotracker green MFI in shCTR and shPERK HeLa cells. Mean±SEM, N=3 biological replicates analyzed using unpaired Student's t test.

(B) Mitochondrial membrane potential measured as TMRM MFI normalized on Mitotracker green MFI in shCTR and shE-Syt1 HeLa cells. Mean \pm SEM, N=3 biological replicates analyzed using unpaired Student's t test.

(C) Percentage of chain length of PC, PI and PE per each PL class from purified mitochondrial fractions of shCTR, shPERK and shE-Syt1 HeLa cells. A representative experiment showing Mean \pm SEM, N=3 technical replicates.

(D) Representative immunoblot for eGFP and ACTIN showing the expression of E-Syt1 transfection in shCTR, shPERK and shPERK+PERK K618A HeLa cells transiently transfected with eGFP-tagged E-Syt1.

(E) Quantification of GFP-E-Syt1 expression normalized on ACTIN. Mean \pm SEM, N=3 biological replicates analyzed using one-way ANOVA, with Tukey's test for multiple comparisons.

Bibliography

1. Kang F, Zhou M, Huang X, Fan J, Wei L, Boulanger J, et al. E-syt1 Re-arranges STIM1 Clusters to Stabilize Ring-shaped ER-PM Contact Sites and Accelerate Ca²⁺ Store Replenishment. *Sci Rep.* 2019 Mar 8;9(1):3975.
2. Saheki Y, Bian X, Schauder CM, Sawaki Y, Surma MA, Klose C, et al. Control of plasma membrane lipid homeostasis by the extended synaptotagmins. *Nat Cell Biol.* 2016 May;18(5):504–515.
3. Giordano F, Saheki Y, Idevall-Hagren O, Colombo SF, Pirruccello M, Milosevic I, et al. PI(4,5)P(2)-dependent and Ca(2+)-regulated ER-PM interactions mediated by the extended synaptotagmins. *Cell.* 2013 Jun 20;153(7):1494–1509.
4. Galmes R, Houcine A, van Vliet AR, Agostinis P, Jackson CL, Giordano F. ORP5/ORP8 localize to endoplasmic reticulum-mitochondria contacts and are involved in mitochondrial function. *EMBO Rep.* 2016 Jun;17(6):800–810.
5. Guyard V, Monteiro-Cardoso VF, Omrane M, Sauvanet C, Houcine A, Boulogne C, et al. ORP5 and ORP8 orchestrate lipid droplet biogenesis and maintenance at ER-mitochondria contact sites. *J Cell Biol.* 2022 Sep 5;221(9).
6. Gurtubay JI. Solubilization of inner mitochondrial membranes by triton X-100. Effect of ionic strength and temperature. *Rev Esp Fisiol.* 1980 Mar;36(1):83–87.
7. Riekhof WR, Voelker DR. Uptake and utilization of lyso-phosphatidylethanolamine by *Saccharomyces cerevisiae*. *J Biol Chem.* 2006 Dec 1;281(48):36588–36596.
8. Riekhof WR, Wu J, Jones JL, Voelker DR. Identification and characterization of the major lysophosphatidylethanolamine acyltransferase in *Saccharomyces cerevisiae*. *J Biol Chem.* 2007 Sep 28;282(39):28344–28352.
9. Jain S, Stanford N, Bhagwat N, Seiler B, Costanzo M, Boone C, et al. Identification of a novel lysophospholipid acyltransferase in *Saccharomyces cerevisiae*. *J Biol Chem.* 2007 Oct 19;282(42):30562–30569.
10. Riekhof WR, Wu J, Gijón MA, Zarini S, Murphy RC, Voelker DR. Lysophosphatidylcholine metabolism in *Saccharomyces cerevisiae*: the role of P-type ATPases in transport and a broad specificity acyltransferase in acylation. *J Biol Chem.* 2007 Dec 21;282(51):36853–36861.
11. Vance JE. Newly made phosphatidylserine and phosphatidylethanolamine are preferentially translocated between rat liver mitochondria and endoplasmic reticulum. *J Biol Chem.* 1991 Jan 5;266(1):89–97.
12. Tasseva G, Bai HD, Davidescu M, Haromy A, Michelakis E, Vance JE. Phosphatidylethanolamine deficiency in Mammalian mitochondria impairs oxidative phosphorylation and alters mitochondrial morphology. *J Biol Chem.* 2013 Feb 8;288(6):4158–4173.

13. Gok MO, Speer NO, Henne WM, Friedman JR. ER-localized phosphatidylethanolamine synthase plays a conserved role in lipid droplet formation. *Mol Biol Cell*. 2021 Nov 24;mbcE21110558T.
14. Gao X, van der Veen JN, Vance JE, Thiesen A. Lack of phosphatidylethanolamine N-methyltransferase alters hepatic phospholipid composition and induces endoplasmic reticulum stress. ... *et Biophysica Acta (BBA)* 2015;
15. Trentzsch M, Nyamugenda E, Miles TK, Griffin H, Russell S, Koss B, et al. Delivery of phosphatidylethanolamine blunts stress in hepatoma cells exposed to elevated palmitate by targeting the endoplasmic reticulum. *Cell Death Discov*. 2020 Feb 18;6:8.
16. Verfaillie T, Rubio N, Garg AD, Bultynck G, Rizzuto R, Decuypere JP, et al. PERK is required at the ER-mitochondrial contact sites to convey apoptosis after ROS-based ER stress. *Cell Death Differ*. 2012 Nov;19(11):1880–1891.
17. Dimmer KS, Rapaport D. Mitochondrial contact sites as platforms for phospholipid exchange. *Biochimica et Biophysica Acta (BBA)-Molecular* 2017;
18. Kuschner CE, Choi J, Yin T, Shinozaki K, Becker LB, Lampe JW, et al. Comparing phospholipid profiles of mitochondria and whole tissue: Higher PUFA content in mitochondria is driven by increased phosphatidylcholine unsaturation. *J Chromatogr B, Analyt Technol Biomed Life Sci*. 2018 Sep 1;1093-1094:147–157.
19. Manders EMM, Verbeek FJ, Aten JA. Measurement of co-localization of objects in dual-colour confocal images. *J Microsc*. 1993 Mar;169(3):375–382.
20. Wieckowski MR, Giorgi C, Lebedzinska M, Duszynski J, Pinton P. Isolation of mitochondria-associated membranes and mitochondria from animal tissues and cells. *Nat Protoc*. 2009 Oct 8;4(11):1582–1590.
21. van Vliet AR, Giordano F, Gerlo S, Segura I, Van Eygen S, Molenberghs G, et al. The ER Stress Sensor PERK Coordinates ER-Plasma Membrane Contact Site Formation through Interaction with Filamin-A and F-Actin Remodeling. *Mol Cell*. 2017 Mar 2;65(5):885–899.e6.

December 27, 2022

RE: JCB Manuscript #202206008R

Prof. Patrizia Agostinis
KU Leuven
Laboratory of Cell Death Research and Therapy - Prof. Patrizia Agostinis - KUL-VIB
Herestraat 49 - O&N1bis - Bus 802
Leuven 3000
Belgium

Dear Prof. Agostinis,

Thank you for submitting your revised manuscript entitled "PERK recruits E-Syt1 at ER-mitochondria contacts for mitochondrial lipid transport and respiration." We would be happy to publish your paper in JCB pending final revisions necessary to meet our formatting guidelines (see details below).

A. MANUSCRIPT ORGANIZATION AND FORMATTING:

1) Text limits: Character count for Articles is < 40,000, not including spaces. Count includes title page, abstract, introduction, results, discussion, and acknowledgments. Count does not include materials and methods, figure legends, references, tables, or supplemental legends.

2) Figure formatting: Articles may have up to 10 main text figures. Scale bars must be present on all microscopy images, including inset magnifications. Molecular weight or nucleic acid size markers must be included on all gel electrophoresis. Please add scale bars to magnifications in Figures 1E/G, 3J, 4G/I, 5C/E, S1B, & S4J.

Also, please avoid pairing red and green for images and graphs to ensure legibility for color-blind readers. If red and green are paired for images, please ensure that the particular red and green hues used in micrographs are distinctive with any of the colorblind types. If not, please modify colors accordingly or provide separate images of the individual channels.

3) Statistical analysis: Error bars on graphic representations of numerical data must be clearly described in the figure legend. The number of independent data points (n) represented in a graph must be indicated in the legend. Please, indicate whether 'n' refers to technical or biological replicates (i.e. number of analyzed cells, samples or animals, number of independent experiments). If independent experiments with multiple biological replicates have been performed, we recommend using distribution-reproducibility SuperPlots (please see Lord et al., JCB 2020) to better display the distribution of the entire dataset, and report statistics (such as means, error bars, and P values) that address the reproducibility of the findings.

Statistical methods should be explained in full in the materials and methods. For figures presenting pooled data the statistical measure should be defined in the figure legends. Please also be sure to indicate the statistical tests used in each of your experiments (both in the figure legend itself and in a separate methods section) as well as the parameters of the test (for example, if you ran a t-test, please indicate if it was one- or two-sided, etc.). Also, if you used parametric tests, please indicate if the data distribution was tested for normality (and if so, how). If not, you must state something to the effect that "Data distribution was assumed to be normal but this was not formally tested."

4) Materials and methods: Should be comprehensive and not simply reference a previous publication for details on how an experiment was performed. Please provide full descriptions (at least in brief) in the text for readers who may not have access to referenced manuscripts. The text should not refer to methods "...as previously described." Please also indicate the acquisition and quantification methods for immunoblotting/western blots.

5) For all cell lines, vectors, constructs/cDNAs, etc. - all genetic material: please include database / vendor ID (e.g., Addgene, ATCC, etc.) or if unavailable, please briefly describe their basic genetic features, even if described in other published work or gifted to you by other investigators (and provide references where appropriate). Please be sure to provide the sequences for all of your oligos: primers, si/shRNA, RNAi, gRNAs, etc. in the materials and methods. You must also indicate in the methods the source, species, and catalog numbers/vendor identifiers (where appropriate) for all of your antibodies, including secondary. If antibodies are not commercial, please add a reference citation if possible.

6) Microscope image acquisition: The following information must be provided about the acquisition and processing of images:

- a. Make and model of microscope
- b. Type, magnification, and numerical aperture of the objective lenses
- c. Temperature
- d. Imaging medium
- e. Fluorochromes
- f. Camera make and model
- g. Acquisition software
- h. Any software used for image processing subsequent to data acquisition. Please include details and types of operations involved (e.g., type of deconvolution, 3D reconstitutions, surface or volume rendering, gamma adjustments, etc.).

7) References: There is no limit to the number of references cited in a manuscript. References should be cited parenthetically in the text by author and year of publication. Abbreviate the names of journals according to PubMed.

8) Supplemental materials: Articles may have up to 5 supplemental figures and 10 videos. Please also note that tables, like figures, should be provided as individual, editable files. A summary of all supplemental material should appear at the end of the Materials and methods section. Please include one brief sentence per item.

9) Video legends: Should describe what is being shown, the cell type or tissue being viewed (including relevant cell treatments, concentration and duration, or transfection), the imaging method (e.g., time-lapse epifluorescence microscopy), what each color represents, how often frames were collected, the frames/second display rate, and the number of any figure that has related video stills or images.

10) eTOC summary: A ~40-50 word summary that describes the context and significance of the findings for a general readership should be included on the title page. The statement should be written in the present tense and refer to the work in the third person. It should begin with "First author name(s) et al..." to match our preferred style.

11) Conflict of interest statement: JCB requires inclusion of a statement in the acknowledgements regarding competing financial interests. If no competing financial interests exist, please include the following statement: "The authors declare no competing financial interests." If competing interests are declared, please follow your statement of these competing interests with the following statement: "The authors declare no further competing financial interests."

12) A separate author contribution section is required following the Acknowledgments in all research manuscripts. All authors should be mentioned and designated by their first and middle initials and full surnames. We encourage use of the CRediT nomenclature (<https://casrai.org/credit/>).

13) ORCID IDs: ORCID IDs are unique identifiers allowing researchers to create a record of their various scholarly contributions in a single place. At resubmission of your final files, please consider providing an ORCID ID for as many contributing authors as possible.

14) Materials and data sharing:

All animal and human studies must be conducted in compliance with relevant local guidelines, such as the US Department of Health and Human Services Guide for the Care and Use of Laboratory Animals or MRC guidelines, and must be approved by the authors' Institutional Review Board(s). A statement to this effect with the name of the approving IRB(s) must be included in the Materials and Methods section.

As a condition of publication, authors must make protocols and unique materials (including, but not limited to, cloned DNAs; antibodies; bacterial, animal, or plant cells; and viruses) described in our published articles freely available upon request by researchers, who may use them in their own laboratory only. All materials must be made available on request and without undue delay. We strongly encourage to deposit all the cell lines/strains and reagents generated in this study in public repositories.

All datasets included in the manuscript must be available from the date of online publication, and the source code for all custom computational methods, apart from commercial software programs, must be made available either in a publicly available database or as supplemental materials hosted on the journal website. Numerous resources exist for data storage and sharing (see Data Deposition: <https://rupress.org/jcb/pages/data-deposition>), and you should choose the most appropriate venue based on your data type and/or community standard. If no appropriate specific database exists, please deposit your data to an appropriate publicly available database. Please, deposit your electron microscopy and mass spectrometry data in appropriate public databases.

15) Please note that JCB now requires authors to submit Source Data used to generate figures containing gels and Western blots with all revised manuscripts. This Source Data consists of fully uncropped and unprocessed images for each gel/blot displayed in the main and supplemental figures. Since your paper includes cropped gel and/or blot images, please be sure to provide one Source Data file for each figure that contains gels and/or blots along with your revised manuscript files. File names

for Source Data figures should be alphanumeric without any spaces or special characters (i.e., SourceDataF#, where F# refers to the associated main figure number or SourceDataFS# for those associated with Supplementary figures). The lanes of the gels/blots should be labeled as they are in the associated figure, the place where cropping was applied should be marked (with a box), and molecular weight/size standards should be labeled wherever possible. Source Data files will be directly linked to specific figures in the published article.

B. FINAL FILES:

Thank you for your attention to these final processing requirements. Please revise and format the manuscript and upload materials within 10 days. If complications arising from measures taken to prevent the spread of COVID-19 will prevent you from meeting this deadline (e.g. if you cannot retrieve necessary files from your laboratory, etc.), please let us know and we can work with you to determine a suitable revision period.

Thank you for this interesting contribution, we look forward to publishing your paper in Journal of Cell Biology.

Sincerely,

William Prinz, PhD
Monitoring Editor
Journal of Cell Biology

Dan Simon, PhD
Scientific Editor
Journal of Cell Biology

Reviewer #1 (Comments to the Authors (Required)):

The authors have responded to my comments and suggestions in full. The study is now a beautiful piece of research for which I commend the authors.

Reviewer #3 (Comments to the Authors (Required)):

The authors answered to all my concerns and the additional work they performed highly improved the quality of the manuscript and clearly demonstrate the role of PERK/SYT1 in the regulation of mitochondrial lipid homeostasis.